# VARIATIONAL REPARAMETRIZED POLICY LEARNING WITH DIFFERENTIABLE PHYSICS

## ABSTRACT

We study the problem of policy parameterization for reinforcement learning (RL) with high-dimensional continuous action space. Our goal is to find a good way to parameterize the policy of continuous RL as a multi-modality distribution. To this end, we propose to treat the continuous RL policy as a generative model over the distribution of optimal trajectories. We use a diffusion process-like strategy to model the policy and derive a novel variational bound which is the optimization objective to learn the policy. To maximize the objective by gradient descent, we introduce the Reparameterized Policy Gradient Theorem. This theorem elegantly connects classical method REINFORCE and trajectory return optimization for computing the gradient of a policy. Moreover, our method enjoys strong exploration ability due to the multi-modality policy parameterization; notably, when a strong differentiable world model presents, our method also enjoys the fast convergence speed of trajectory optimization. We evaluate our method on numerical problems and manipulation tasks within a differentiable simulator. Qualitative results show its ability to capture the multi-modality distribution of optimal trajectories, and quantitative results show that it can avoid local optima and outperforms baseline approaches.

## 1 INTRODUCTION

Reinforcement learning (RL) with *high-dimensional continuous action space* is notoriously hard despite its fundamental importance for many application problems such as robotic manipulation (OpenAI et al., 2019; Mu et al., 2021). Compared with the discrete action space counterpart setup, it is much trickier to represent policies for continuous RL – by the optimality theory of RL, the function class of discrete RL policies is simply categorical distributions (Sutton & Barto, 2018), while the function class of continuous RL policies has to include density functions of arbitrary probabilistic distributions.

In practice, popular frameworks (Silver et al., 2014; Haarnoja et al., 2018; Schulman et al., 2017) of deep RL formulate the continuous policy as a neural network that outputs a single-modality density function over the action space (e.g., a Gaussian distribution over actions). This formulation, however, breaks the promise of RL being a global optimizer of the return function because the single-modality policy parameterization introduces local minima that are hard to escape using gradients w.r.t. distribution parameters. Besides, a single-modality policy will significantly weaken the exploration ability of RL algorithms because the sampled actions are usually concentrated around the modality. Our Bandit examples show how a single-modality RL policy fails to solve a simple continuous action bandit problem. Therefore, in practice, continuous RL often requires meticulous reward design that takes considerable human effort (Mu et al., 2021; Savva et al., 2019; Yu et al., 2020; Makoviychuk et al., 2021; Zhu et al., 2020).

In this paper, we propose a principled framework to learn the continuous RL policy as a multi-modality density function. We provide a holistic solution to two closely-related questions: *1) how to parameterize the continuous RL policy? 2) how to update the policy parameterized as in (1)?*

Our parameterization of the continuous RL policy is based on two ideas. First, we take a sequence modeling perspective (Chen et al., 2021) and view the policy as a density function over *the entire trajectory space* (instead of the action space) (Ziebart, 2010; Levine, 2018). Under this sequence modeling perspective, we can sample a population of trajectories that cover multiple modalities of

trajectories, which allows us to concurrently explore distant regions in the solution space. Second, we use a generative model to parameterize the multi-modality policies, inspired by their success in modeling highly complicated distributions such as natural images (Goodfellow et al., 2016; Zhu et al., 2017; Rombach et al., 2022; Ramesh et al., 2021). We introduce a sequence of latent variables $z$, and we learn a decoder that "reparameterizes" the random distribution $z$ into the multi-modality trajectory distribution (Kingma & Welling, 2013), from which we can sample trajectories $\tau$. Our policy network is, in a spirit, akin to diffusion models (Rombach et al., 2022; Ramesh et al., 2021) by learning to model the joint distribution $p(z, \tau)$. As a prototypical work, we prefer simple design and choose not to include the multi-step diffusion process as in image modeling.

Our choice of policy parameterization leads us to adopt the variational method (Kingma & Welling, 2013; Haarnoja et al., 2018; Moon, 1996) to derive an on-policy learning algorithm. Classical on-policy learning leverages the policy gradient theorem (Sutton & Barto, 2018), i.e., $\nabla J(\pi) = \mathbb{E}_\tau[R(\tau)\nabla \log p(\tau)]$. Because we model $p(z, \tau)$, that requires computing $p(\tau)$ and its gradient with $\int_z p(z, \tau)\,\mathrm{d}z$. However, marginalizing out $z$ is often intractable when $z$ is continuous, and it is well-known that optimizing the marginal distribution $\log p(\tau)$ by gradient descent suffers from local optimality issues (e.g., using gradient descent to optimize Gaussian mixture models which have latent variables is not effective, so EM is often used instead Ng (2000)). To overcome these obstacles, we take a different route and adopt variational method (maximum entropy RL) that directly optimizes the joint distribution of the optimal policy without hassels of integrating over $z$.

We derive a novel variational bound which is the optimization objective for policy learning. To maximize this objective, we introduce the **Reparameterized Policy Gradient Theorem**. The theorem states a principled way to compute the policy gradient by combining the reward-weighted gradient (as in the classical policy gradient theorem) and the path-wise gradient from a differentiable world model (as in classical trajectory optimization methods). The two sources of gradient complement each other – the reward-weighted gradient improves the likelihood of selecting trajectories from regions containing high-reward ones, whereas the path-wise gradient suggests updates to make local improvements to trajectories. This combination allows us to enjoy the precise gradient computation from differentiable world models (Werling et al., 2021; Hu et al., 2019; Huang et al., 2021) and also maintains the flexibility to sample and optimize the trajectory distribution globally. Note that this beautiful result is also a consequence of introducing the latent variable $z$.

The effectiveness of our method is rooted in our choice of generative policy modeling method. An ideal method needs to be powerful in modeling multi-modality distribution, and it needs to support sampling, density value computation, and stable gradient computation. Although there are other candidates for deep generative models, they often have limitations to being used for continuous policy modeling. For example, GAN-like generative models (Goodfellow et al., 2020) can only sample but not compute the density value. While normalizing flow (Rezende & Mohamed, 2015) can compute the density value, they might not be as robust numerically due to the dependency on the determinant of the network Jacobian; moreover, normalizing flows must apply continuous transformations onto a continuously connected distribution, making it difficult to model disconnected modes (Rasul et al., 2021).

We apply our methods to several numerical problems and three trajectory optimization tasks to manipulate rigid-body objects and deformable objects supported by differentiable physics engines (Werling et al., 2021; Huang et al., 2020). These environments contain various local optima that challenge single-modality policies (either reward-weighted policy gradients as in REINFORCE or path-wise gradients as in trajectory optimization). In contrast, our approach benefits from modeling the trajectories as generative models and the ability to sample in the trajectory space. In qualitative experiments, we observe a strong pattern of multi-modality distribution by visualizing the policy after the learning converges. By quantitative evaluation, the policy learned by our framework does not suffer from the local optimality issue and significantly outperforms baselines.

## 2 VARIATIONAL REPARAMETERIZED POLICY LEARNING

In this section, we introduce our basic framework. All notations follow the convention of the community. To be more clear, we leave a section that introduces background knowledge in Appendix A and a mathematical table for notations in Appendix B.

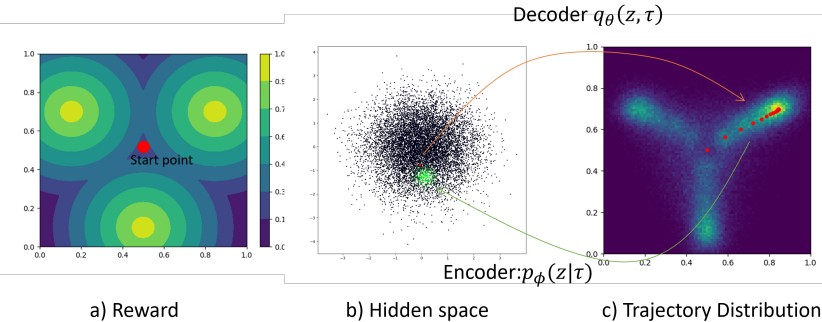

Figure 1: a) the reward landscape where the agent needs to move from the red dots to the region containing a high reward; b) latent space $\mathcal{Z}$ modeled by random Gaussian; c) the state density of a sampled trajectory from $q_\theta(z, \tau)$. In (c), each red dot corresponds to a state in the trajectory. Our policy can be viewed as encoding the stochastic latent variable $z$ into the trajectory distribution through the decoder $q_\theta(z, \tau)$. We rely on an encoder $p_{\phi(z|\tau)}$ to ensure the cycle consistency between the latent $z$ and the sampled trajectory $\tau$.

## 2.1 GENERATIVE MODELING OF OPTIMAL TRAJECTORIES

Following (Todorov, 2006; 2008; Toussaint, 2009; Ziebart, 2010; Kappen et al., 2012; Levine, 2018; Haarnoja et al., 2018), our Variational Reparameterized Policy (VRP) framework, as shown in Fig. 1, views policy optimization as learning a generative model that generates optimal trajectories. To capture the multimodalities of the optimal trajectories, we introduce a latent distribution and construct a decoder-encoder structure similar to recent deep generative models (Kingma & Welling, 2013; Ho et al., 2020; Rombach et al., 2022; Ramesh et al., 2021). The "encoder" maps trajectories to latent samples, while the decoder is a neural network that "reparameterizes" random samples from the latent distribution into different trajectories.

**Method Overview.**    As introduced in Sec. 2.2, the latent distribution and the reparameterization function together form a policy that can generate diverse trajectories. We use a novel variational bound to approximate the posterior of optimal trajectories in Sec. 2.3 as the optimization objective. The variational bound naturally combines maximum entropy RL while also containing a term to enforce cycle consistency (Zhu et al., 2017) between the encoder and the decoder, preventing the policy from mode collapse. We then introduce a Reparameterized Policy Gradient Theorem in Sec. 2.4 to optimize our reparameterized policy to improve the variational bound. We find that the derived policy gradient includes two terms: one optimizes the latent distribution through a reward-weighted gradient as the classical policy gradient theorem, and another optimizes the reparameterization network, which can benefit from the path-wise gradient from a differentiable world model. This helps our inference algorithm to enjoy the efficiency of differentiable physics while being able to sample globally when the reward landscape is discontinuous or non-convex.

## 2.2 TRAJECTORY GENERATION WITH REPARAMETERIZED POLICY

Let $z \in \mathcal{Z}$ be a latent variable, which can be either continuous or categorical to model optimal trajectories. We design our "policy" as a joint distribution $q_\theta(z, \tau)$ of latent $z$ and the trajectory $\tau$. To model a sequential trajectory, we can consider the following two factorizations: 1. sample $z$ before $\tau$: $q_\theta(z, \tau) = p(s_1)q_\theta(z|s_1) \prod_{t=1}^{T} p(s_{t+1}|s_t, a_t)q_\theta(a_t|z, s_t)$; 2. sample $z$ with $\tau$: $q_\theta(z, \tau) = p(s_1)q_\theta(z_0|s_1) \prod_{t=1}^{T} p(s_{t+1}|s_t, a_t)q_\theta(a_t|z_{t-1}, s_t)q_\theta(z_t|s_t, z_{t-1})$. The latter allows us to model the hybrid policy as in the option-critic (Bacon et al., 2017), where we can treat $z_t$ as the option per step.

Though sampling from $q_\theta$ may require us to sample $\tau, z$ together, it is still convenient to view it as the combination of two marginal distributions: (1) the policy for the latent representation $p_\theta(z|s_1)p(s_1)$, and (2) a decoder $p_\theta(\tau|z, s_1)$ that reparameterizes $z$ into a real trajectory. Note that we will use $p_\theta$ to refer to various marginal distribution of $q_\theta(z, \tau)$. For example, $p_\theta(\tau) = \int_z q_\theta(z, \tau)dz$ will be the marginal distribution of trajectories sampled from $q_\theta(z, \tau)$.

The term "reparameterization" is inspired by the reparameterization trick in (Kingma & Welling, 2013). When $z$ is a Gaussian noise $\xi$, we can reparameterize the noise into actions by $a = \mu(\theta) + \xi\sigma(\theta)$ and optimize it with gradients directly. The reparameterized policy $q_\theta$ can be viewed as an instance of the stochastic computation graph (Schulman et al., 2015; Weber et al., 2019). In practice, $q_\theta(z|s)$ or $q_\theta(a|s,z)$ are modeled by neural networks as in common policy learning frameworks (Haarnoja et al., 2018).

## 2.3 THE VARIATIONAL LOWER BOUND FOR REPARAMETERIZED POLICIES

**The Auxiliary Trajectory Encoder**  Intuitively, learning policy networks by inputting random distribution can help generate diverse examples, but it will also suffer from the issues of mode collapse Li et al. (2017). To build a connection between the trajectory and $\tau$ and enforce a cycle consistency, we introduce an auxiliary distribution $p_\phi(z|\tau)$, which can look at the whole trajectories and map $\tau$ back into the latent $z$. We will illustrate how the auxiliary encoder naturally emerges from the variational lower bounds following (Rezende & Mohamed, 2015; Levine, 2018).

**The Evidence Lower Bound**  With the help of the auxiliary encoder $p_\phi(z|\tau)$, we can now define a joint distribution of optimality $O$, latent $z$, and the trajectory $\tau$ as $p_\phi(O, z, \tau) = p(O|\tau)p_\phi(z|\tau)p(\tau)$ where $O$ and $z$ are independent conditioning on $\tau$. Treating $q_\theta(z, \tau)$ as the posterior approximation, we can write the Evidence Lower Bound (ELBO) for the optimality. Note that the following inequality holds for arbitrary distribution $p_\phi$ and $q_\theta$,

$$\log p(O) = \underbrace{\mathbb{E}_{z,\tau\sim q_\theta}\left[\log p_\phi(O, z, \tau) - \log q_\theta(z, \tau)\right]}_{\text{ELBO}} + D_{KL}(q_\theta(z,\tau)||p_\phi(z,\tau|O))$$

$$\geq \mathbb{E}_{z,\tau\sim q_\theta}\left[\log p_\phi(O, \tau, z) - \log q_\theta(z, \tau)\right]$$

$$= \mathbb{E}_{z,\tau\sim q_\theta}\left[\log p(O, \tau) + \log p_\phi(z|\tau) - \log q_\theta(z, \tau)\right]$$

$$= \mathbb{E}_{z,\tau\sim q_\theta}\left[\underbrace{\log p(O|\tau)}_{\text{reward}} + \underbrace{\log p(\tau)}_{\text{prior}} + \underbrace{\log p_\phi(z|\tau)}_{\text{cross entropy}} - \underbrace{\log q_\theta(z, \tau)}_{\text{entropy}}\right] \quad (1)$$

The ELBO contains four parts that can all be computed directly given the sampled $z$ and $\tau$ (the environment probability $p(s_{t+1}|s_t, a_t)$ is canceled as in (Levine, 2018)). The first two parts are the predefined reward $\log p(O|\tau) = R(\tau)/\mathcal{T} + c$, where $c$ is the normalizing constant that can be ignored during optimization, and a prior distribution $p(\tau)$, which is assumed to be known. The third part is the log-likelihood of $z$ based on our encoder. It is easy to see that if we fix $q_\theta$, and maximize $p_\phi$ alone will minimize the cross-entropy $\mathbb{E}_{z,\tau\sim q_\theta}[-\log p_\phi(z|\tau)]$, similarly to the supervised learning. It achieves optimality when $p_\phi(z|\tau) = p_\theta(z|\tau) = \frac{q_\theta(z,\tau)}{\int_z q_\theta(z,\tau)dz}$, modeling the posterior of $z$ for $\tau$ sampled from $q_\theta$. On the other hand, fixing $\phi$, the decoder $q_\theta$ is encouraged to generate trajectories that are easy to identify or classify; this helps to increase diversity and enforce a cycle consistent to avoid mode collapse. The fourth part is the policy entropy that enables maximum entropy exploration.

Maximizing all terms together for the parameters $\theta$ and $\phi$ will minimize

$$D_{KL}(q_\theta(z,\tau)||p_\phi(z,\tau|O)) = D_{KL}(q_\theta(z,\tau)||p_\phi(z|\tau)p(\tau|O)),$$

where optimality can be achieved when $p_\theta(z|\tau) = p_\phi(z|\tau)$ and $p_\theta(\tau)p_\theta(z|\tau) = p_\phi(z|\tau)p(\tau|O) \Rightarrow p_\theta(\tau) = \int q_\theta(\tau, z)dz = p(\tau|O)$. We discuss the method's connection with other methods in Appendix C.

## 2.4 REPARAMETERIZED POLICY GRADIENT WITH DIFFERENTIABLE PHYSICS

Given $\theta, \phi$ we can treat the ELBO as our reward $R_{\text{elbo}}(\tau) = \log p(O|\tau) + \log p(\tau) + \log p_\phi(z|\tau) - \log q_\theta(z, \tau)$. The maximization w.r.t. $\phi$ is straightforward. As for $q_\theta$, depending on its structure, we can optimize it with various on-policy or off-policy RL methods as in (Weber et al., 2019).

Here we study a special case where $R_{\text{elbo}}(\tau)$ is differentiable to $\theta$ through a sampled trajectory $\tau$. Thus, we can optimize $\theta$ by first-order path-wise gradients directly. This case is very interesting as analytical gradients provide a fast convergence speed but also suffer from the local optima issue (Li

et al., 2017), while our reparameterization policy naturally includes a latent distribution for sampling and a parameterization part for optimization, providing more chances to search globally. One can obtain a path-wise gradient from a differentiable simulator (Hu et al., 2019; Werling et al., 2021), a learned neural model (Liang et al., 2022).

**Reparameterized Policy in Differentiable Environments.** Formally, we hope to find $\theta$ to maximize the expectation $\mathbb{E}[R_{\text{elbo}}(\tau)] = \int_{z,\tau} q_\theta(z,\tau) R_{\text{elbo}}(\tau) d\tau dz$[1]. The sampling procedure is usually non-differentiable. But if we let $a_t = \mathbf{f}_\theta(s_t, z, t)$ and the dynamics $s_{t+1} = h(s_t, a_t)$ be differentiable functions, the trajectory $\tau$ becomes a differentiable function of $\theta$ almost everywhere.

We further assume that $s_1$ is fixed and the functions $\mathbf{f}_\theta, h$ are deterministic to simplify the derivation. One can remove these assumptions easily through reparameterization as in (Silver et al., 2014; Heess et al., 2015) to generalize to stochastic functions. As a result, we can write $q_\theta(z, \tau_\theta) = q_\theta(z|\tau(z, s_1))$, where $\tau$ and the density $q_\theta$ is a deterministic function of the sampled $z$ and $s_1$. In this case, $\mathbb{E}[R_{\text{elbo}}(\tau_\theta)] = \int_z q_\theta(z, \tau_\theta) R_{\text{elbo}}(\tau_\theta) dz$. For each sampled trajectory $\tau_\theta$, we can compute its path-wise gradient w.r.t. to the policy parameter $\theta$ as $\nabla_\theta R_{\text{elbo}}(\tau_\theta)$. Then we have the following theorem to compute the gradient of the expected rewards:

**Theorem 1** (Reparameterized Policy Gradient Theorem)**.** *Given $s_1$ and under regularity conditions in Appendix F.1. For almost every $\theta$, the expected reward $E[R_{elbo}(\tau_\theta)]$ exists and differentiable, and its gradient can be computed by*

$$\nabla_\theta \mathbb{E}[R_{elbo}(\tau_\theta)] = \int_z q_\theta(z, \tau_\theta) \left[ \underbrace{R_{elbo}(\tau_\theta) \nabla_\theta \log q_\theta(z, \tau_\theta)}_{\substack{\textit{Reward-weighted Gradient} \\ \textit{(REINFORCE)}}} + \underbrace{\nabla_\theta R_{elbo}(\tau_\theta)}_{\substack{\textit{Pathwise Gradient} \\ \textit{(Trajectory Optimization)}}} \right] dz.$$

We provide a short proof in Appendix F.2.

**Combine Sampling and Optimization.** Our theorem shares the same spirit as in Schulman et al. (2015) and is related to the problem of exchange derivative and expectation L'Ecuyer (1995). Here we want to emphasize its use case in trajectory optimization. Theorem 1 splits gradients into two parts. The first part is a reward-weighted gradient that aims to improve the likelihood of the distribution $q_\theta(z|\tau) = q_\theta(z, \tau_\theta)$ for high-reward ones. If the environment is differentiable, the second part may optimize the policy directly through a path-wise gradient to make local improvements to trajectories. The natural combination of the sampling and optimization may allow our approach to enjoy the benefits of both: it can search over the whole space but can also leverage local structures for fast convergence speed. This can not be achieved without the latent variable $z$.

Our methods provide the potential to avoid discontinuities in the reward landscape. If $q_\theta(z, \tau_\theta)$ does not depend on $s_{>1}$, the reward-weighted gradient can optimize the sampling distribution directly without relying on the path-wise gradients, providing the policy a chance to ignore the discontinuous point and move to the high reward region directly. Another appealing point in Theorem 1 is that in Assumption 2, we only require $R_{\text{elbo}}(\tau_\theta)$ to be Lipschitz continuous for $q_\theta(z, \tau_\theta) \geq 0$. Thus, our policy gradient estimate can make an unbiased estimation so long as the probability density of sampling a discontinuous trajectory is negligible. Even if a discontinuous point generates a biased gradient estimation for a sampled $z$, the zeroth-order part still has a chance to correct its density directly by observing the reward directly so long as the optimal trajectories have certain properties.

By defining latent distributions, building the reparameterized policy and the auxiliary encoder, then optimizing the ELBO with the reparameterized policy gradient through a differentiable world model, we can combine search and optimization to enjoy the generative modeling of the optimal trajectories. We summarize the whole recipe in Algorithm 1 of Appendix D and describe implementation details in Appendix D.

---

[1] Here we assume $z$ contains only continuous variables to simplify the derivation. Adding discrete variables is straightforward by summing over all possibilities.

## 3 RELATED WORK

**Differentiable Simulation and Trajectory Optimization.** Trajectory optimization (Kelly, 2017) aims to find a trajectory which optimizes the target metrics under given constraints over the trajectory. One branch of methods optimizes the trajectory using the gradient of the optimization objective (Ratliff et al., 2009; Kalakrishnan et al., 2011; Schulman et al., 2014). Recently, the development of differentiable simulation technique (Hu et al., 2019; Werling et al., 2021; Qiao et al., 2021; Freeman et al., 2021; Howell et al., 2022) enables to compute the gradient from the reward function and optimize the trajectory in physical simulators. Such methods are efficient around the optimal solution; however, the optimization often gets stuck at local optima elsewhere, especially when the reward function is not smooth. In our work, one term in the formula of our RPG theorem corresponds to the gradient from trajectory optimization.

**Policy as Sequential Generative Model.** Maximum entropy reinforcement learning (Todorov, 2006; 2008; Toussaint, 2009; Ziebart, 2010; Kappen et al., 2012) can be viewed as variational inference in probabilistic graphical models (Levine, 2018), which models optimality as observed variable and trajectory as a latent variable. When the demonstration or a fixed dataset is provided on the *offline* RL setting (Chen et al., 2021; Reed et al., 2022), policy learning is simplified as a sequence modeling task (Chen et al., 2021; Zheng et al.; Reed et al., 2022). They use autoregressive models to learn the distribution of the whole trajectory, including actions, states, and rewards, and use the action prediction as policy. In our work, we learn a sequential generative model of policy for *online* RL via the variational method. The policy can model any distributions in the trajectory space.

**Variational Skill Discovery** Our method is closely related to the work in unsupervised reinforcement learning (Eysenbach et al., 2018; Achiam et al., 2018) or diverse skill learning (Kumar et al., 2020; Osa et al., 2022). These methods share the same technique of using neural networks to approximate the posterior of the latent variable given either states or state-action pairs and encourage the policy to reach states consistent with the latent variables. However, these methods do not model the optimal trajectory distribution but only aim to generate a diverse set of solutions by adding the mutual information term as a reward bonus, resulting in different formulations and effects. For example, they all choose to fix initial latent distributions without optimizing them, limiting their ability to achieve optimality. Moreover, Eysenbach et al. (2018); Achiam et al. (2018) does not optimize the learned skill for the environment rewards; Osa et al. (2022) does not optimize the mutual information along trajectories; Kumar et al. (2020) needs to solve the optimization problem first before finding a diverse set of solutions. In contrast, we jointly optimize the latent representation and the policy with a single objective, providing a simple but unified perspective for previous approaches in optimization problems.

**Hierarchical Methods** As mentioned in Sec. 2.2, the hierarchical methods, e.g., option critic (Bacon et al., 2017), can be seen as a special case of our method when we use a sequence of latent variables $z = (z_1, \cdots, z_T)$ to reparameterize the policy. Without optimizing the latent variable through the variational inference, most hierarchical RL methods often need special designs for the latent space, e.g., state-based subgoals (Kulkarni et al., 2016; Nachum et al., 2018b;a) or predefined skills (Li et al., 2020) to avoid mode-collapse. Osa et al. (2019) regularized options to maximize the mutual information between the action and the options, which are very relevant to ours. However, it does not model temporal structures as ours to ensure consistency along the trajectories. Hierarchical imitation learning Gupta et al. (2019); Pertsch et al. (2021); Shankar & Gupta (2020); Jiang et al. (2022); Lynch et al. (2020); Fang et al. (2020) extract temporal abstractions using generative models from demonstrations. InfoGAN (Li et al., 2017) and ASE (Peng et al., 2022) uses adversarial training Goodfellow et al. (2020); Ho & Ermon (2016) to imitate demonstrations. These works all rely on demonstrations rather than rewards to learn abstractions. For example, (Co-Reyes et al., 2018) learns representation on the collected dataset with variational inference and then utilizes the trained model for planning or policy learning. The separation of the representation learning and reward maximization makes it differ from our methods: first, it requires a state reconstruction module to supervise the generative model, which is challenging for high-dimensional observations; second, it optimizes neither the latent distribution nor the actions for the reward directly, thus requires additional planning procedure during the execution to find suitable actions.

## 4 EXPERIMENT

### 4.1 NUMERICAL EXPERIMENTS

#### 4.1.1 ENVIRONMENT AND EVALUATION

In this section, we investigate the following questions: 1) Whether our method is more efficient for trajectory optimization than gradient-free algorithms; 2) Whether our method can search over the large solution space and avoid local optima that might trap a single-modality policy.

We design the following environments to answer these questions. **Bandit 1 and 2**: Our bandit problems in Figure 2a-(1) and Figure 2a-(2) have a 1d action space and a non-convex reward landscape. We initialize our policy as a Gaussian centralized around $0$ with its scale barely touching the right mode of reward. The reward function of *Bandit 2* contains an additional discontinuous point. **Move 1 and 2**: An agent moves in a 2D environment for a fixed number of steps and receives a reward according to its terminal position. The terminal reward landscape contains 4 Gaussian peaks (deeper color = higher reward), as shown in Figure 2a-(3) and (4). The agent is initialized at the center. Note that the reward of *Move 1* near the initialization point is constant, so the agent receives a non-zero gradient only if it arrives at a position closer to one of the four Gaussian peaks. **Move 3**: The terminal reward shown in Figure 2a-(5). There is one obstacle consisting of three white circles. When an agent runs into an obstacle, it will bounce off (continuous collision detection (Hu et al., 2019)) due to an impulse normal to the contact surface. Note that there is a small dent between the three circles, creating a local optimum.

We evaluate our method against the single modality policies learned from the following baselines: 1) Reinforce (Williams, 1992), and 2) policies optimized through path-wise gradients with Adam (Kingma & Ba, 2014). We compare their sample efficiency and final performance after training is finished.

#### 4.1.2 EXPERIMENT RESULTS

We plot the average episode return of each algorithm against the number of samples in Figure 2b. Error bars show the standard deviations over five runs with different random seeds. The results suggest that our methods achieve better sample efficiency and find better solutions than baseline algorithms.

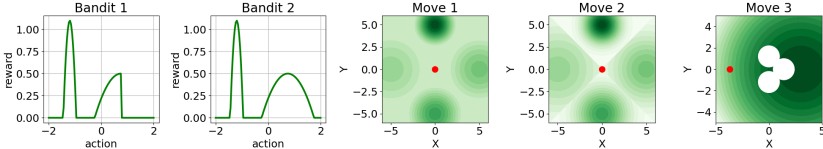

(a) Environments for numerical experiments. We plot the reward landscapes in the first two bandit environments and show the rewards of the terminating positions as contour plots in the right three 2d move environments. These environments all pose some level of challenge in optimizing reward/value landscapes that are discontinuous and/or locally optimal.

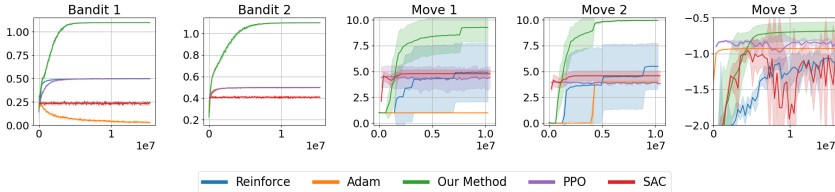

(b) Training curve. X-axis: number of sampled states. Y-axis: average return. Error bars show standard deviations over 5 runs. Hyperparameters are available in Table 2.

We first take **Bandit 2** in Figure 2a-(2), which has two modes and regions containing zero gradients in its reward landscape, as an example for comparison. We show the performance of each algorithm in Figure 2b-(2). In terms of optimality, our method converges to the globally optimal solution in the end and achieves a much higher return compared with other methods in Figure 2b-(2). Two reasons

prevent our method from being stuck at the local optima. First, the entropy term in the $R_{\text{elbo}}$ objective encourages the agent to perform a maximum entropy exploration over the whole reward landscape, while the cross entropy term drives the agent to sample distinguishable actions for different latent variables, causing different action modes determined by latent variables to move away from each other, resulting in a better state coverage. Second, after discovering different local optima, our policy is able to maintain a multi-modality action distribution to exploit those local optima independently. By adjusting the probabilities of different modes through optimizing rewards with latent variables, it can eventually converge to the global optimal solution. In experiments, we observed that although our method quickly found the right mode of the reward function, shown in Figure 2a-(2), which is locally optimal, it would still increase the probabilities for the under-explored region due to the exploration terms in our objective. After it found the left mode, it would maintain two action modes and optimize them together until the expected return of the left mode was larger than the right one and eventually converged to the left, finding the global optimum.

A notable feature of the multi-modality is that during the whole optimization process, the expected reward increases monotonically as it only needs to increase the reward for modes with higher rewards. In contrast, when Adam and Reinforce converge to the right mode, even if we have introduced the maximum entropy reward to encourage exploration. different from our method, their single modality property prevents them from jumping from one mode to a better mode as there is no way to go across the flat region in between without decreasing the reward.

In terms of sample efficiency, our method benefits from the pathwise analytical gradient, consistently outperforming Reinforce, which has a noisy gradient estimate that may reduce convergence speed when the number of samples is limited. Though Adam increases rewards faster than our method in the beginning stage, it quickly gets stuck at the local optima. Current RL algorithms like PPO or SAC are also constrained by the single modality gaussian parameterization of the policy, these methods can only find suboptimal solutions, even when more than enough samples are given.

**Remaining environments** Experiments on the remaining four environments show a similar behavior pattern that our method can converge to the global optima that the baselines fail to find. We observe that the curve of ADAM in Figure 2b-(1) even decreases after reaching a local optimum. This is due to the discontinuity on the right of the reward landscape as shown in Figure 2a-(1). When a Gaussian policy reaches the right local optimum, the analytical gradients nearing that point still only contain a positive value, pushing the policy to go across the optimal point and move into the right flat region, causing the return to decrease to $-0.5$ in the end. Reinforce and PPO does not suffer from this issue, as it only uses the reward-weighted gradient estimate, and can correctly identify the left low-reward region and move away from it. SAC also does not suffer from this issue, as the policy optimizes the Q function, which smooth the discontinuity to allow the policy to converge at the local maximum stably. But it is still very challenging to find the global optima with a similar reason as in the **Bandit 2** environment.

In the case of sequential decision-making problems, our method consistently outperforms algorithms that maintain single modality policy in several **move** environments by a large margin and can find global optima as shown in Figure 2b (3)-(5). We observed that Reinforce went to the left side mode in **Move 1** and **Move 2** as at the starting state there are no reward signals pointing to the global optimum in the upper side of the state space. Its variance is also high due to the increased problem dimensions. Adam even did not make any progress in **Move 1** as there is no gradient around the initial state. The **Move 3** environment includes physics-based contacts in the dynamic system, while our method still can solve it with the analytical gradients, in presence of the discontinuities during contacts.

### 4.1.3 HOW LATENT DISTRIBUTIONS AFFECT THE TRAJECTORY MODELING?

Modeling the trajectories with a sequential $\{z_i\}$ instead of a single latent variable sampled before sampling the trajectory $\tau$ can help factorize the trajectory space to build a more compact representation. We construct an environment containing four obstacles to demonstrate its effects in Figure 3. The agent on the bottom left must avoid obstacles and reach the goal at the top right corner. We study the case where $z$ is a discrete variable. We first plot the state and the action distributions for policies learned by sampling a single categorical $z$ before the trajectory $\tau$ in Figure 3a. Different colors represent the different values of $z$. We plot the state and action distributions and use different colors to represent their corresponding $z$ values. Though a categorical distribution can represent various paths toward the goal, the number of latent samples constrained its expressive power. However, as shown in

Figure 3b, we model the distribution with a sequence of $z_1, z_{K+1}, z_{2K+1}, \ldots, z_{NK+1}$. This brings the compositionality that allows the agents to select various $z$ sequences to generate trajectories that can better cover the state space. The action distribution on the right successfully captures the symmetry of the optimal trajectories and the action pattern of moving along different directions, which helps the policy better cover the whole state space compared with a single categorical distribution.

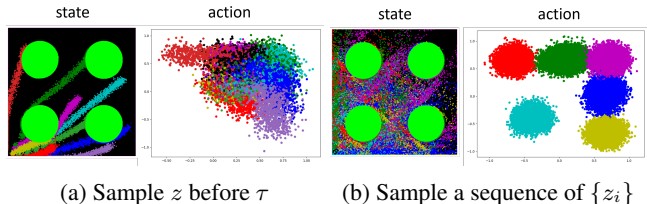

(a) Sample $z$ before $\tau$      (b) Sample a sequence of $\{z_i\}$

Figure 3: A navigation task to demonstrate the effects of learning a sequence of $z$. An agent needs to move from the bottom left to the top right and avoid the green obstacles. Both the state and action space are 2D. Different colors represent different values of $z$ or $z_i$.

## 4.2 EVALUATION ON MANIPULATION TASKS WITH DIFFERENTIABLE PHYSICS

We evaluate our approach in several physical environments to demonstrate its potential in trajectory optimization. Figure 4 compares our approaches against the gradient-based trajectory optimizer with a single Gaussian head in three environments. The **Grasp** environment, based on Nimble Physics (Werling et al., 2021), contains three balls. The agent needs to control a 3 DoF gripper to grasp the green ball and pick it up. However, the agent only knows that it needs to grasp an object, and it does not receive any reward before lifting it. We use a reward to minimize the distance between the gripper and the nearest balls, which creates a *local minimum* that encourages the gripper to touch the nearest ball in front of it, as shown in Figure 4. The first-order gradient-based trajectory optimization got stuck. In contrast, our method can capture the multimodality of the reward landscape, approach different balls, and exploit different options (touching different balls) simultaneously. In the end, our method successfully finds the solution to pick up the green ball. The **Rope** and the **Cut** environments are two soft body environments built with PlasticineLab (Huang et al., 2020), each containing a 3DoF rigid body manipulator. In the **Rope** environment, an agent needs to push the left side of a rope forward. Similarly, in the **Cut** environment, an agent needs to cut off the left end of a rope. However, manipulators are initialized in the middle of the objects, which is far away from suitable contact points required to finish the tasks, leading to local optima (Li et al., 2022) as shown in Fig. 4. In contrast, our method can explore different contact points and find correct solutions in the end. These experiments show the potential of our approach in learning the optimal policies with differentiable physics.

## 5 CONCLUSION AND FUTURE WORK

We derive a framework that models the policy of continuous RL by a multi-modality distribution in the variational inference framework. Under this framework, we also derive a Reparameterized Policy Gradient Theorem which enjoys the advantage of both classical sample-based methods (as in REINFORCE) and trajectory optimization methods (which require the support of differentiable world models).

Our framework opens a new venue of continuous RL. It has deep connections with diffusion model, decision sequence modeling, and differentiable physics techniques. In the future, we are interested in exploring how sequence modeling techniques, such as transformers, can be used to model the policy in our framework. We are also interested in testing our method for more complicated tasks, such as dexterous object manipulation. Finally, our method can be extended to offline RL and off-policy RL setups. We leave all these exploration efforts to future works.

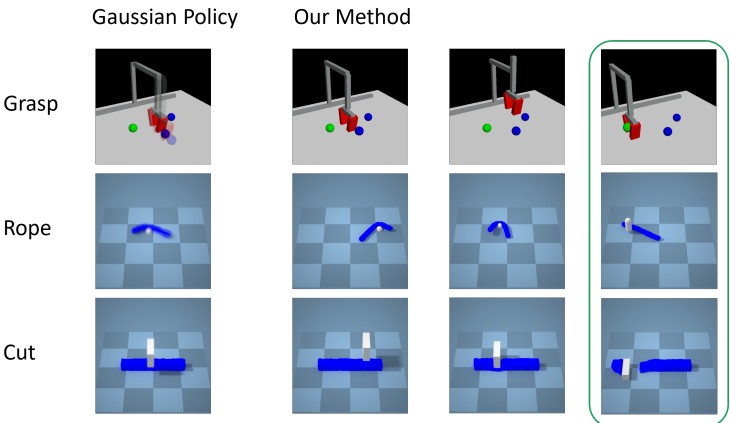

Figure 4: Visualization of policy distribution for manipulation with differentiable physics. We qualitatively compare the Gaussian policy learned by ADAM and the multi-modality policy learned by our method. On the left, we overlay final states from three trajectories of the Gaussian policy. On the right, we also draw three samples (trajectories) from the policy, and we visualize the final state of each trajectory in a different plot. We see that the three final states from the Gaussian policy are similar, while the final states are very different for trajectories from our policy. Additionally, one of the modalities of our method can solve the problem (the rightmost column), while the Gaussian policies get stuck in local minima.

## REPRODUCIBLITY STATEMENT

We will provide an open-source implementation of our method on GitHub. We will also update more illustrative examples and demos to our website. The hyperparameter used in the experiment section can be found in Table 2 in the appendix. The theoretical results are also in multiple sections of our appendix.

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

## A PRELIMINARY

**Markov decision process**    A Markov decision process (MDP) is a tuple of $(\mathcal{S}, \mathcal{A}, \mathcal{P}, \mathcal{R})$, where $\mathcal{S}$ is the state space and $\mathcal{A}$ is the action space. $p(s'|s, a)$ is the transition probability that transits state $s$ to another state $s'$ after taking action $a$. The function $R(s, a, s')$ computes a reward per transition. A policy $\pi(a|s)$ selects an action distribution according to the state $s$. Executing a policy $\pi$ starting from the initial state $s_1$ with density $p(s_1)$ will result in a *trajectory* $\tau$, which is a sequence of states and actions $\{s_1, a_1, s_2, \ldots, s_t, a_t, \ldots\}$ where $a_t \sim \pi(a|s = s_t), s_{t+1} \sim p(s|s = s_t, a = a_t)$. We also use the terminology *environment* to refer to an MDP in an RL problem. The discounted reward of a trajectory is $R_\gamma(\tau) = \sum_{t=1}^{\infty} \gamma^t R(s_t, a_t, s_{t+1})$ where $0 < \gamma < 1$ is the discount factor to ensure the series converges. The goal of reinforcement learning (RL) is to find a parameterized policy $\pi_\theta$ that maximizes the expected reward $E_{s_1 \sim p(s_1)} V^{\pi_\theta}(s_1) = E_{\tau \sim \pi_\theta, s_1 \sim p(s_1)}[R_\gamma(\tau)]$, where we call $V^{\pi_\theta}(s_1)$ the value of the state $s_1$ for the policy $\pi_\theta$.

For simplicity, we focus on optimization problems in a finite horizon MDP in this paper. This means that we only consider the first $T$ states and actions $\{s_1, a_1, \ldots, s_T, a_T, s_{T+1}\}$. In this case, we can directly optimize for the total reward $R(\tau) = \sum_{t=1}^{T} R(s_t, a_t, s_{t+1})$. Defining the measure and ensuring the existence of policy gradients for infinite horizon MDP with differentiable physics requires more strict conditions, and we leave it as future work.

**Policy gradient and zeroth-order gradient estimate**    REINFORCE (Sutton & Barto, 2018) approximates the expected reward with Monte-Carlo sampling and optimizes the policy parameter $\theta$ with a zeroth order gradient estimate, which can be written as $\frac{\partial E_{s_1 \sim p(s_1)}[V^{\pi_\theta}(s_1)]}{\partial \theta} \approx \sum_{i=1}^{N_{\text{samples}}} R(\tau^i) \nabla_\theta \log \pi_\theta(\mathbf{a}^i)$ where $N_{\text{samples}}$ is the number of sampled trajectories, $\tau^i$ and $\mathbf{a}^i = \{a_t^i\}_{1 \le t \le T}$ represent the trajectory and action sequence per sample, and the log-likelihood can be computed by $\log \pi_\theta(\mathbf{a}^i) = \sum_{1 \le t \le T} \log \pi_\theta(a_t|s_t)$.

**Differentiable simulation**    In differentiable physics we consider as a special MDP where the transition probability $p(s'|s, a)$ can be represented as $\delta(s' - h(s, a))^2$. The transition function $h : \mathcal{S} \times \mathcal{A} \to \mathcal{S}$ and reward function $R(s, a, s')$ are deterministic and differentiable. If we further assume that the policy $\pi_\theta(a|s) = \delta(a - \mathbf{f}_\theta(s))$ is a differentiable deterministic function w.r.t. $\theta$ and input $s$, then each element in a sampled trajectory $\tau$ and the total reward $R(\tau)$ are also differentiable w.r.t. each $s_t, a_t$ and $\theta$. We can optimize the reward $R(\tau)$ with gradient-based optimizers after computing the first-order path-wise gradient $\frac{\partial R(\tau)}{\partial \theta}$ through back-propogation.

It's argued that even if the environment is deterministic, it is still beneficial to optimize a stochastic policy (Suh et al., 2022; Xu et al., 2022) to leverage the advantages of stochastic sampling for the non-smooth reward landscape (Hu et al., 2019; Xu et al., 2022; Antonova et al., 2022). A typical choice is to add Gaussian noise $\mathbf{w}$ to the action sequence $\mathbf{a}$ and optimize it with the reparameterization trick (Kingma & Welling, 2013), which estimates the same policy gradient as REINFORCE by $\frac{\partial E_{s_1 \sim p(s_1)}[V^{\pi_\theta}(s_1)]}{\partial \theta} \approx \sum_{i=1}^{N_{\text{samples}}} \nabla_\theta R(\tau^i(\mathbf{a}^i + \mathbf{w}^i))$, where $\mathbf{a}^i$ and $\mathbf{w}^i$ are actions and Gaussian noises to generate the $i$-th trajectory $\tau^i$.

**RL as Probabilistic Inference**    The RL as inference framework (Todorov, 2006; 2008; Toussaint, 2009; Ziebart, 2010; Kappen et al., 2012; Levine, 2018), which defines optimality $p(O|\tau) \propto e^{R(\tau)/\mathcal{T}}$, where $\mathcal{T}$ is a defined temperature. It further defines a prior distribution of the trajectory $p(\tau) = p(s_1) \prod_{t=1}^{T} p(a_t|s_t) p(s_{t+1}|s_t, a_t)$, where $p(a_t|s_t)$ is a known prior action distribution, e.g., a Gaussian distribution. Thus, we can compute the possibility of the optimality $p(O) = p(\tau|O)p(\tau)$. The goal of the framework is to approximate the posterior distribution of optimal trajectories $p(\tau|O) = \frac{p(O|\tau)p(\tau)}{\int p(O|\tau)p(\tau)d\tau}$, which can be considered as a softmax weighting among all prior trajectories based on their rewards, sharing the same spirit to path integral and the SoftMax policy.

---

$^2\delta$ is the Dirac delta function.

## B  MATHEMATICAL NOTATIONS

| Notation | Explanation |
|---|---|
| $z$ | The latent variable |
| $\tau$ | The sampled trajectory |
| $p(O\|\tau)$ | Optimality of a trajectory $\tau$ |
| $p(\tau)$ | Prior distribution of trajectories (for example, random actions) |
| $p(O)$ | $\int p(O\|\tau)p(\tau)d\tau$. |
| $p(\tau\|O)$ | The posterior policy we want to model |
| $p_\phi(z\|\tau)$ | The introduced auxiliary encoder |
| $q_\theta(z,\tau)$ | Joint distribution of $z,\tau$ by sampling from the reparameterized policy in the environment $p(s_{t+1}\|s_t,a_t)$. |
| $q_\theta(a_t\|s_t,z)$ or $q_\theta(a_t\|s_t,z_t)$ | The policy to select actions at different steps |
| $q_\theta(z\|s_1)$ or $q_\theta(z_t\|s_t,z_{t-1})$ | The policy to sample latent $z$ |
| $p_\theta(z\|\tau)$ | The probability of sampling $z$ or all $z_i$ given $\tau$ in $q_\theta$ |
| $p_\theta(z\|s_1)$ | The marginal distribution of $z$ in $q_\theta$ conditioning on $s_1$ |
| $p_\theta(\tau)$ | The marginal distribution of $\tau$ in $q_\theta$ |
| $p_\theta(\tau\|z)$ | The marginal distribution of $\tau$ conditioning on $z$ in $q_\theta$ |
| $R_{\text{elbo}}$ | The variational lower bound defined in Eq. 1 |
| $\mathbf{f}_\theta(s_t,z,t)$ | The deterministic policy function. |
| $h(s_t,a_t)$ | The deterministic state transition function. |
| $q_\theta(z\|\tau)$ | Usually the $q_\theta(z,\tau_\theta)$ in deterministic environments |
| $\tau_\theta$ or $\tau_\theta(z,s_1)$ | In deterministic environment, the trajectory becomes a function of $\theta$ and $s_1$ |
| $\nabla_\theta R_{\text{elbo}}(\tau_\theta)$ | The path-wise gradient obtained from the differentiable environment |

## C  CONNECTION WITH GENERATIVE MODELS

Many generative models are based on ELBO

$$\log p(x) = E_{z\sim q(z)}\left[\log p(x,z) - \log q(z)\right] + KL(q(z)\|p(z|x)).$$

One can compare our approaches with other generative models. VAE defines $p_\theta(x,z) = p_\theta(x|z)p(z)$ and $q(z) = q_\phi(z|x)$ and then optimize $\theta,\phi$ jointly to maximize the ELBO bound. By doing so, $q_\phi(z|x)$ has to align with the true posterior of $p_\theta(z|x)$. Thus

$$\log p(x) \geq E_{z\sim q_\phi(z|x)}[\log p_\theta(x|z) + \log p(z) - \log q_\phi(z|x)]$$

The Expectation–maximization algorithm (EM) (Dempster et al., 1977) assumes that we have $p_\theta(x,z) = p_\theta(x|z)p_\theta(z)$ with finite $z$, thus we can compute the expectation $p_\theta(z|x)$ explicitly and treat it as $q_\phi$, then E-step: compute $p_\theta(z|x)$ to solve $\max_\phi \log p_\theta(x) - D_{KL}(q_\phi(z|x)\|p_\theta(z|x))$. M-step: fixing $\phi$, find $\max_\theta E_{q_\phi}[\log p_\theta(x,z)] - E_{q_\phi}[\log q_\phi(z|x)]$ can maximize the ELBO bound.

In Maximum Entropy RL, we have $p(O,\tau) = p(O|\tau)p(\tau)$ defined by the reward, and we optimize $q_\theta(\tau|O)$ only. The ELBO bound becomes a maximum entropy term $\mathbb{E}_{\tau\sim\pi}\left[\log p(O|\tau) + \log p(\tau) - \log \pi(\tau)\right]$.

| | Latent | Encoder $q(z\|x)$ | Joint $p(x,z)$ | MLE objective |
|---|---|---|---|---|
| VAE | $z$ | $p_\phi(z\|x)$ | $p_\theta(x\|z)p(z)$ | $p(x)$ |
| EM | $z$ | $\max_\phi \log p_\theta(x) - D_{KL}(q_\phi(z\|x)\|p_\theta(z\|x))$ | $p_\theta(x\|z)p_\theta(z)$ | $p(x)$ |
| Diffusion | $\{x_t\}_{t\geq 1}$ | $\prod_{i=1}^T \mathcal{N}(x_t; \sqrt{1-\beta_t}x_{t-1}, \beta_t\mathbf{I})$ | $p(x_T)\prod_{t>1} p_\theta(x_{t-1}\|x_t)$ | $p(x_0)$ |
| MaxEntRL | $\tau$ | $\pi_\theta(\tau)$ | $p(O\|\tau)p(\tau)$ | $p(O)$ |
| RPG | $\tau, z$ | $q_\theta(z,\tau)$ | $p(O\|\tau)p_\phi(z\|\tau)p(\tau)$ | $p(O)$ |

Table 1: Comparison of different algorithms that optimize ELBO bounds for inference

## D  IMPLEMENTATION DETAILS

The algorithm is shown in Algorithm 1.

---

**Algorithm 1** Variational Reparameterized Policy

---

1: **Input:** $p_\phi, q_\theta(z, \tau)$.
2: Initialize $p_\phi, q_\theta$.
3: **while** time remains **do**
4:     Sample start state $s_1$.
5:     Sample from the latent code policy $z, \tau$.
6:     Compute the variational lower bound $R_{\text{elbo}}$ based Eq. 1.
7:     Update the auxiliary encoder $p_\phi$ to maximize $R_{\text{elbo}}$.
8:     Update the parameterized policy gradient $q_\theta$ to maximize $R_{\text{elbo}}$ with Theorem 1.
9: **end while**

---

Table 2: Hyperparameters and rewards of our algorithms.

| Hyperparameter\Env | Bandit1 | Bandit 2 | Move1 | Move2 | Move3 |
|---|---|---|---|---|---|
| environment steps | 1 | 1 | 8 | 8 | 50 |
| num epoch | 50 | 50 | 50 | 50 | 50 |
| batch size | 1000 | 1000 | 256 | 256 | 500 |
| samples per epoch | 50000 | 50000 | 12800 | 12800 | 25000 |
| batch size | 1000 | 1000 | 256 | 256 | 500 |

The learning rate $(3 \times 10^{-4})$ is the same across all environments. Adam uses the exact same hyperparameter as our method except for the dimension of the latent variable is 1. REINFORCE uses the exact same hyperparameter as our method except for the dimension of the latent variable is 1 and the policy gradient is estimated using reinforce.

# E BIAS OF THE FIRST-ORDER GRADIENT ESTIMATOR

In this section, we analyze the difference between two types of policy gradient estimators, namely the Zeroth-Order (ZoBG) and First-Order (FoBG) gradient estimator (Suh et al., 2022). We establish a necessary condition for two to be equivalent when both action and parameter are 1 dimensional. From the 1D case, we also show the intractability of constructing a gradient estimator similarly to these two in higher dimensions.

**Lemma 1** (Leibniz integral rule, general form). *Let $f(x, t)$ be a function such that both $f(x, t)$ and its partial derivative $\frac{\partial f(x,t)}{\partial x}$ are continuous in t and x in some region of the $xt$ plane, including $a(x) \leq t \leq b(x)$, $x_0 \leq x \leq x_1$. Also suppose that the functions $a(x)$ and $b(x)$ are both continuous and both have continuous derivatives for $x_0 \leq x \leq x_1$. Then, for $x_0 \leq x \leq x1$*

$$\frac{d}{dx} \int_{a(x)}^{b(x)} f(x,t)dt = \int_{a(x)}^{b(x)} \frac{\partial}{\partial x} f(x,t)dt + \frac{d}{dx}b(x) \cdot f(x, b(x)) - \frac{d}{dx}a(x) \cdot f(x, a(x))$$

**Lemma 2** (Jump Discontinuity, 1D version). *Let f(x, t) be a function such that both $f(x, t)$ and its partial derivative $\frac{\partial f(x,t)}{\partial x}$ are continuous in t and x in some region of the $xt$ plane. f has jump discontinuity if and only if there exist $b_i = q_i(x)$ such that*

$$\lim_{t \to q_i(x)^-} f(x,t) \neq \lim_{t \to q_i(x)^+} f(x,t)$$

We define $i$ to correspond the ith jump discontinuity of $f$. For convenience, we also abbreviate the above as $f(b_i^-) \neq f(b_i^+)$ in later parts of this section.

**Corollary 1** (Leibniz integral rule, 1D region form). *Let f(x, t) be a function such that both $f(x, t)$ and its partial derivative $\frac{\partial f(x,t)}{\partial x}$ are continuous in t and x in some region of the $xt$ plane, including*

$t \in T, T = \mathbb{R}$, $x \in X, X = \mathbb{R}$. *Also suppose that the functions $q_i(x)$ for all $i$ are continuous and have continuous derivatives for $x \in \mathbb{R}$. Then, by Lemma 1 and 2, for $x \in X$*

$$\frac{d}{dx} \int_T f(x,t)dt = \underbrace{\int_T \frac{\partial}{\partial x} f(x,t)dt}_{\text{interior part}} + \underbrace{\sum_i \frac{\partial q_i(x)}{\partial x} \left( f(x, q_i(x)^-) - f(x, q_i(x)^+) \right)}_{\text{boundary part}}$$

For convenience, we also refer to the first term as the interior part and the second term as the boundary part in the later parts of this section.

**Theorem 2** (Policy Gradient Theorem, zeroth order version). *Let $\pi_\theta(a)$ be a function such that $\pi_\theta(\cdot) \in C^0$ for all $\theta \in \mathbb{R}$ and $\pi_{(\cdot)}(a) \in C^1$ for all $a \in A$, $A = \mathbb{R}$. Let $R(a)$ be a function with jump discontinuities and defined for all $a \in A$*

$$\frac{d}{d\theta} \int_A \pi_\theta(a)R(a)da = \int_A \frac{\partial \pi_\theta(a)}{\partial \theta} R(a)da$$

*holds.*

*Proof.* Define the integrand $\pi_\theta(a)R(a) = f(\theta, a)$, by Corollary 1

$$\frac{d}{d\theta} \int_A \pi_\theta(a)R(a)da$$
$$= \int_A \frac{\partial \pi_\theta(a)}{\partial \theta} R(a)da + \sum_i \frac{\partial q_i(\theta)}{\partial \theta} \left( f(\theta, q_i(\theta)^-) - f(\theta, q_i(\theta)^+) \right)$$

Case 1: If $\pi_\theta(b_i) \neq 0$, by Lemma 2, the definition of the ith jump discontinuity can be rewritten as

$$q_i(\theta) = b_i : f(\theta, b_i^-) \neq f(\theta, b_i^+)$$
$$: \pi_\theta(b_i^-)R(b_i^-) \neq \pi_\theta(b_i^+)R(b_i^+)$$
$$: R(b_i^-) \neq R(b_i^+) \qquad\qquad \pi_\theta(\cdot) \in C^0 \text{ for all } \theta$$

, which means $q_i(\theta)$ is constant with respect to $\theta$. $\frac{\partial q_i(\theta)}{\partial \theta} = 0$ reduces the boundary part to 0 and gives

$$\frac{d}{d\theta} \int_A \pi_\theta(a)R(a)da = \int_A \frac{\partial \pi_\theta(a)}{\partial \theta} R(a)da + 0$$

Case 2: If $\pi_\theta(b) = 0$, by Lemma 2, then $f(\theta, b) = \pi_\theta(b)R(b) = 0$, which means $f(\theta, b^-) = f(\theta, b^+) = 0$ and $f$ does not have jump discontinuity at $b$. □

**Lemma 3** (Law of the unconscious statistician, continuous random variable). *Let $x$ be a random variable and let $y = g(x)$ be a function of this random variable.*

$$E[g(x)] = \int_X f_X(x)g(x)dx$$

**Theorem 3** (Policy Gradient Theorem, first-order version). *Let $\omega$ be a random variable such that $a = h(\theta, \omega)$ with probability distribution $p(\omega) \in C^0$ and $\frac{\partial h(\theta, \omega)}{\partial \omega} \neq 0$ for all $\theta$. Let $f(\theta, \omega) = p(\omega)R(h(\theta, \omega))$ with the ith jump discontinuity of $f$ defined to be $q_i(\theta)$. The first order gradient is*

$$\frac{d}{d\theta} \int_A \pi_\theta(a)R(a)da = \int_\Omega p(\omega)\frac{\partial}{\partial \theta} R(h(\theta, \omega))d\omega + \sum_i \frac{\partial q_i(\theta)}{\partial \theta} \left( f(\theta, q_i(\theta)^-) - f(\theta, q_i(\theta)^+) \right)$$

*Proof.* Reparameterize $a = h(\theta, \omega)$, substitute in $f$ by definition

$$\frac{d}{d\theta} \int_A \pi_\theta(a)R(a)da = \frac{d}{d\theta} \int_\Omega p(\omega)R(h(\theta, \omega))d\omega = \frac{d}{d\theta} \int_\Omega f(\theta, \omega)d\omega$$

By Lemma 2 the definition of the ith jump discontinuity can be rewritten as

$$q_i(\theta) = b_i : f(\theta, b_i^-) \neq f(\theta, b_i^+)$$
$$: p(b_i^-)R(h(\theta, b_i^-)) \neq p(b_i^+)R(h(\theta, b_i^+))$$
$$: R(h(\theta, b_i^-)) \neq R(h(\theta, b_i^+)) \qquad\qquad p(\omega) \in C^0$$

Suppose $\omega \sim \mathcal{N}(0, \sigma)$, $h(\theta, \omega) = \theta + \sigma\omega$ and exist action $a_i = h(\theta, b_i)$ such that $R(a_i^-) \neq R(a_i^+)$, the definition of $q_i(\theta)$ and its evaluation at a local neighbour $q_i(\theta + \Delta\theta)$ can be rewritten as

$$q_i(\theta) = b_i : h(\theta, b_i) = a_i$$
$$q_i(\theta + \Delta\theta) = b_i^\Delta : h(\theta + \Delta\theta, b_i^\Delta) = a_i$$

since function R is not parameterized, the location of its ith jump discontinuity is $a_i$ and does not change. Therefore, substitute in $h(\theta, \omega) = \theta + \sigma\omega$ and solve for $b_i$ and $b_i^\Delta$ gives $b_i = (a_i - \theta)/\sigma$ and $b_i^\Delta = (a_i - (\theta + \Delta\theta))/\sigma$. Differentiate $q_i(\theta)$ gives

$$\frac{dq_i(\theta)}{d\theta} = \lim_{\Delta\theta \to 0} \frac{q_i(\theta + \Delta\theta) - q_i(\theta)}{\Delta\theta}$$
$$= \lim_{\Delta\theta \to 0} \frac{b_i^\Delta - b_i}{\Delta\theta}$$
$$= \lim_{\Delta\theta \to 0} \frac{\Delta\theta/\sigma}{\Delta\theta}$$
$$= \frac{1}{\sigma} \neq 0$$

In general, ignoring the boundary term when estimating the left-hand side with a first-order estimator leads to a bias due to the discontinuity of the reward/value function. □

Therefore, we have shown that a gradient estimator for only the interior part of the first-order gradient is a biased estimator of the original policy gradient. The bias is the boundary term.

# F    Proof of the Reparameterized Policy Gradient Theorem

## F.1    Regularity Conditions

Let $\theta \in \Theta = (-\epsilon, \epsilon) \in \mathbb{R}$ without loss of generality and $\Omega = \{z \in \mathcal{Z} | q_\theta(z, \tau_\theta) > 0\}$ be a measure space, then for $z \in \Omega, \theta \in [-\epsilon, \epsilon]$, we make the following assumptions

**Assumption 1** (Bounded ELBO). *The reward R, prior density $\log p(\tau), \log q_\theta(z|\tau), \log p_\phi(z|\tau)$ are bounded for all $\tau$.*

**Assumption 2** (Lipschitz Condition). *$q_\theta(z|\tau)$, $\log q_\theta(z|\tau), \log p_\phi(z|\tau)$, prior distribution $\log p(\tau)$, action policy $\mathbf{f}_\theta(s_t, z, t)$, reward R and dynamics $h(a_t, s_t)$ are Lipschitz continuous w.r.t. any $s, a$ and continuous $z$. In addition, $q_\theta$ is Lipschitz continuous w.r.t. $\theta$.*

**Assumption 3** (Lebesgue-integrable). *The probability density $q_\theta(z, \tau_\theta)$ and its derivative $|\nabla_\theta q_\theta(z, \tau_\theta)|$ is Lebesgue-integrable jointly over $\Omega$ and their integration is bounded for all $\theta \in [-\epsilon, \epsilon]$.*

Assumption 1 is easy to guarantee when the state space and the action space are compact and the time horizon $T$ is finite because continuous functions on a compact set are always bounded. Similarly, for bounded inputs, the common neural networks can also satisfy Assumption 2. Otherwise, we can simply clamp them before feeding them into a neural network to make the input bounded, as we do not need the network and the environment to be continuously differentiable as in Lemma 1 but only absolutely continuous. We require the dynamics $h(a_t, s_t)$ to be Lipschitz continuous, ruling out chaotic systems.

Assumption 3 holds for common distributions. For example, given a Gaussian distribution $p(z) = e^{(-z^2/(2\sigma^2))}/(\sqrt{2\pi}\sigma)$, the parameter $\sigma$'s partial derivative $\frac{\partial p}{\partial \sigma}(z) = e^{(-z^2/(2\sigma^2))}(-\sigma^2 + z^2)/(\sqrt{2\pi}\sigma^4)$ is simply a function of exponential family and can be verified to be absolutely integrable on $\mathbb{R}$ and bounded for any domain $\sigma \in [a, b], a, b > 0$.

## F.2 PROOF OF THEOREM 1

To differentiate under integration, we use the following Lemma (cheng; L'Ecuyer, 1995),

**Lemma 4** (Leibniz integral rule). *Let $X$ be an open subset of $\mathbb{R}$, and $\Omega$ be a measure space. Suppose that a function $f : X \times \Omega \to \mathbb{R}$ satisfies the following conditions:*

1. *$f(x, \omega)$ is a measurable function of $x$ and $\omega$ jointly and is integrable over $\omega$ for almost all $x \in X$ fixed.*

2. *For almost all $\omega \in \Omega$, $f(x, \omega)$ is an absolutely continuous function of $x$ (the derivative $\partial f(x, \omega)/\partial x$ exists almost everywhere).*

3. *$\partial f/\partial x$ is "locally integrable", that is, for all compact intevals $[a, b] \in X$,*

$$\int_a^b \int_\Omega \left| \frac{\partial}{\partial x} f(x, \omega) \right| d\omega dx < \infty.$$

*Then for almost every $x \in X$, its derivative exists and*

$$\frac{d}{dx} \int_\Omega f(x, \omega) d\omega = \int_\Omega \frac{\partial}{\partial x} f(x, \omega) d\omega$$

Then it is easy to prove Theorem 1.

*Proof.* Define the measurable function $f(\theta, z) = q_\theta(z, \tau_\theta) R_{\text{elbo}}(\tau_\theta)$.

Since $R_{\text{elbo}}(\tau_\theta)$ is bounded by a constant $M$ by Assumption 1 and $q_\theta(z, \tau_\theta)$ is positive and Lebesgue-integrable by Assumption 3, their product's absolute value will be bounded by $q_\theta(z, \tau_\theta) M$, so $f(\theta, z)$ becomes Lebesgue-integrable for every $\theta$.

By Assumption 2 and assuming the horizon is finite, the trajectory $\tau$ will become a Lipschitz function with respect to $\theta$ and $z$, so as $R_{\text{elbo}}(\tau_\theta)$, $q_\theta(z, \tau_\theta)$ and their products $f(\theta, z)$. Thus $\frac{\partial f(\theta, z)}{\partial \theta}$ exists almost everywhere for all $z \in \Omega$.

To bound its derivative, let $R_{\text{elbo}}(\tau_\theta)$ be $L$-Lipschitz w.r.t. $\theta$, notice that

$$|\nabla_\theta f(\theta, z)| = |R_{\text{elbo}}(\tau_\theta) \nabla_\theta q_\theta(z, \tau_\theta) + q_\theta(z, \tau_\theta) \nabla_\theta R_{\text{elbo}}(\tau_\theta)| \le M |\nabla_\theta q_\theta(z, \tau_\theta)| + L |q_\theta(z, \tau_\theta)|,$$

is dominated by the sum of two Lebesgue-integrable functions whose integration is bounded for all $\theta \in \Theta$ by Assumption 3. Thus its gradient must be locally integrable. Then applying Lemma 4, we can do differentiation under the integration sign and finish the proof as

$$\nabla_\theta E[R_{\text{elbo}}(\tau_\theta)] = \int_z \nabla_\theta \left( q_\theta(z, \tau_\theta) R_{\text{elbo}}(\tau_\theta) \right) dz$$

$$= \int_z R_{\text{elbo}}(\tau_\theta) \nabla_\theta q_\theta(z, \tau_\theta)(z|s_1) + q_\theta(z, \tau_\theta) \nabla_\theta R_{\text{elbo}}(\tau_\theta) dz$$

$$= \int_z q_\theta(z, \tau_\theta) \left[ R_{\text{elbo}}(\tau_\theta) \nabla_\theta \log q_\theta(z, \tau_\theta) + \nabla_\theta R_{\text{elbo}}(\tau_\theta) \right] dz.$$

$\square$

We want to emphasize the importance of the Lipschitz continuity that helps to bound the integration. Informally, the Lipschitz condition is directly related to the scale of the path-wise gradient. When the Lipschitz constant is too large, the environments will generate exploding gradients or contains sharp changes in the reward landscape, similarly to the discontinuity and causing empirical bias Suh et al. (2022). Besides, when the reward landscape has discontinuous points, there will be a bias caused by the motion of the discontinuity, as suggested by the Leibniz integral rule. We refer interested readers to Appendix E for a detailed illustration of the difference between the zeroth-order gradient estimate and the first-order gradient estimate under the expectation we do not find in the literature. Note that we do not ask for the continuity of the gradients and support activation functions such as ReLU.

The finite horizon assumption is also necessary to make the theorem holds, otherwise the gradients $\nabla_\theta R_{\text{elbo}}(\tau_\theta)$ may be unbounded if we do not set up a suitable discount factor.

# G LEARNING A DIFFERENTIABLE WORLD MODEL

## G.1 METHOD

To apply our method in a non-differentiable environment, we train a differentiable world model jointly with the policy optimization as Model-based Reparameterized Policy Gradient (MBRPG). This method has been proven data-efficient for policy optimization (Hafner et al., 2019; Schrittwieser et al., 2020; Ye et al., 2021; Hansen et al., 2022).

Specifically, besides of the encoder $p_\phi(z|s, a)$, the policy $\pi_\theta(a|s, z)$, $\pi_\theta(z|s_1)$ we additionally learn a deterministic dynamic network $h_\psi(s, a)$, a reward network $R_\psi(s, a)$, an observation encoder $f_\psi(o)$ and the $Q-$network $Q_\psi(s, a, z)$. In practice, they are all two-layer fully connected neural networks with a hidden dimension 256. We also define $\alpha, \beta$ to weigh the entropy term and the cross entropy term in the variational lower bound.

Given any $z$ and any initial latent state $s_i = f_\psi(o_i)$, and an arbitrary action sequence, we can use the learned dynamic network to generate the trajectory by

$$
\begin{aligned}
a_t &\sim \pi_\theta(a|s_t, z) \text{ or } a_t = a_t^{gt} && \text{(policy)} \\
r_t &= R_\psi(s_t, a_t) && \text{(reward)} \\
r_t' &= -\alpha \log \pi_\theta(a_t|s_t, z) + \beta \log p_\phi(z|s_t, a_t) && \text{(intrinsic reward)} \\
Q_t &= Q_\psi(s_t, a_t, z) && \text{(Q value)} \\
s_{t+1} &= h_\psi(s_t, a_t) && \text{(dynamics)}
\end{aligned}
$$

for $t = i, i+1, \ldots, i+K$. Notice that during the model rollout, we can either use the action $a_t$ sampled from the current policy $\pi_\theta(a|s_t, z)$ or an action sequence $\{a_t^{gt}\}$ sampled from the replay buffer. When the action is sampled from the current policy $\pi_\theta(a|s_t, z)$, we obtain a Monte-Carlo estimate for the value of $s_i$, which can be used to optimize the policy $\pi_\theta$:

$$
V_{\text{estimate}}(o_i, z) \approx \gamma^K (Q_{i+K} + r_{i+K}') + \sum_{t=i}^{i+K-1} \gamma^j (r_t + r_t') \tag{1}
$$

To train the dynamics model, we sample trajectory segments of length $K + 1$ $\tau_{i:i+K} = \{o_i, a_i^{gt}, r_i^{gt}, o_{i+1}, a_{i+1}^{gt}, r_{i+1}^{gt}, \ldots, o_{i+K}\}$ from the replay buffer and select a latent $z$. We then self-supervise the dynamic network to ensure state consistency and avoid reconstruction as in (Ye et al., 2021; Hansen et al., 2022) (in this case we let $a_t = a_t^{gt}$ in the trajectory):

$$
L_\psi(\tau) = \sum_{t=i}^{i+K-1} L_1 \|s_{t+1} - \mathbf{ng}(f_\psi(o_{t+1}))\|^2 + L_2(r_t - r_t^{gt})^2 + L_3(Q_t - \mathbf{ng}(r_t^{gt} + \gamma V_{\text{estimate}}(o_{t+1}, z)))^2 \tag{2}
$$

where $\mathbf{ng}(x)$ means stopping gradient and $L_1 = 1000, L_2 = L_3 = 0.5$ are constant to balance the loss. The whole process is illustrated in Algorithm 2. For all experiments, we take $K = 6$.

## G.2 EXPERIMENT

We first compare our method with SAC Haarnoja et al. (2018) on locomotion tasks (Cheetah-v3 and Humanoid-v3 in OpenAI Gym Brockman et al. (2016)) in Figure 6(a) and (b). We plot the learning curve of MBRPG together with the SAC's performance after training for 3 million steps. We can see that by learning the model, our method achieves on-par performance with fewer samples. In particular, our method only requires 1 million samples to reach 15000 scores on Cheetah-v3 and only 0.5 million samples to reach 6000+ scores on Humanoid-v3. This attributes to the efficiency brought by learning a model for policy optimization. However, we observed that removing the latent variables and the variational bound, which is a baseline model-based RL algorithm (MBRL), did not affect the performance of our method. We conjecture that these locomotion tasks do not require a multi-modal policy for exploration, and a Gaussian policy is sufficient.

To illustrate the effectiveness of our approach, we have built two environments that need multi-modality explorations, as illustrated in Figure 5. In (a) AntMove environments, an ant robot can move

---

**Algorithm 2** Model-based Variational Reparameterized Policy

---

1: **Input:** $p_\phi, \pi_\theta, h_\psi, R_\psi, f_\psi, Q_\psi$
2: **while** time remains **do**
3:     Sample start state $o_1$ and encode it as $f_\psi(o_1)$. Select $z$ from $\pi_\theta(z|s_1)$.
4:     Execute the policy $\pi(a|s, z)$ and store transitions into the replay buffer $\mathcal{B}$.
5:     Sample a batch of trajectory segment of length $K$ $\{\tau_{t:t+K}^i, z\}$ from the buffer $\mathcal{B}$.
6:     Optimize $\psi$ using Equation 2.
7:     Optimize $\pi_\theta(a|s, z)$ with gradient descent to maxmize the value estimate in Equation 1 for $s, z$ sampled from the buffer.
8:     Optimize $\pi_\theta(z|s_1)$ with policy gradient to maximize $V_{\text{estimate}}(s_1, z) - \alpha \log \pi_\theta(z|s_1)$ for $s_1$ is sampled from the buffer.
9:     Optimize $\phi, \alpha, \beta$ and other auxiliary networks.
10: **end while**

---

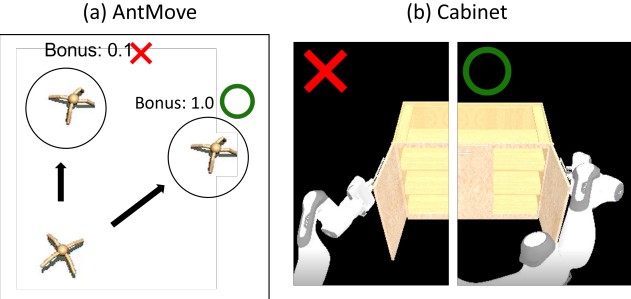

Figure 5: Non-differentiable environments

either upward or rightward to reach one of the two goal regions represented by the black circle. The agent always receives a penalty proportional to its distance to the closest goal, which motivates it to go upward instead of moving right. However, if the agent chooses the longer right path, it will receive a higher bonus when it reaches the goal. Such a local optimum will trap a single modality agent, as shown in Figure 6 (c). When we remove the latent space and the latent policy $p(z|s_1)$, the baseline single-modal RL policy (MBRL) will get stuck in the local optima, while our approach (MBRPG), thanks to its ability to maintain multi-modality trajectory distribution, will attempt to go right even when it is sub-optimal at the beginning and thus has a higher chance to find the global optima. We also test our method for an $11 - dof$ mobile robot for opening the cabinet shown in Figure 5 (b). It is easier for the robot to open the left door, but it will receive a higher reward when it opens the right one. Similarly, the normal RL agent modeled by a Gaussian distribution (MBRL) fails to explore a way to open the right door. In contrast, our method explores the two directions simultaneously and can open the right one, in the end, resulting in higher rewards, as shown in Figure 6.

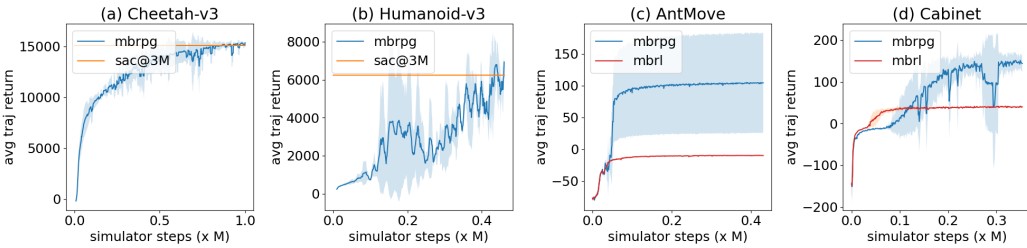

Figure 6: Experiments with a learned world model.

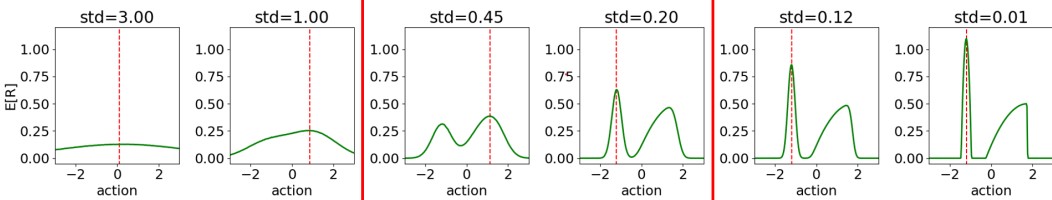

Figure 7: The landscape of expected reward after a gaussian filter with gradually decreasing standard deviation.

## H INSUFFICIENCY OF THE SINGLE MODALITY GAUSSIAN POLICY

Our method is motivated by the insufficiency of the Gaussian policy in solving non-convex problems. A gaussian policy with large entropy has proven effective in many continuous control problems. However, even if the Gaussian distribution has a very high initial variance, it may still fail to solve many tasks. The Gaussian parameterization can only model single-modality action distribution, which limits the policy update in both REINFORCE and ADA. It would prevent them from converging to the global optima even though it can explore the whole reward landscape.

Without loss of generality, let us take the 1d continuous bandit problem as an example. In this case, a Gaussian policy is fully determined by its mean $\mu$ and variance $\sigma^2$. Both REINFORCE and ADAM maximize the expected reward $E_{a \sim \mathcal{N}(\mu,\sigma)}[R(a)]$ by computing its policy gradient $\nabla E_{a \sim \mathcal{N}(\mu,\sigma)}[R(a)]$. We now assume $R(a)$ is Lipschitz continuous and differentiable almost everywhere. As we show in the paper, the gradients computed by the two methods are the same.

If the standard deviation $\sigma$ is not large enough, a Gaussian policy will suffer from the non-convexity issue like a deterministic optimizer. It is easy to see that $\nabla_\mu E_{a \sim \mathcal{N}(\mu,\sigma)}[R(a)] = \nabla_\mu \int_a 1/(\sigma\sqrt{2\pi})e^{-\frac{(\mu-a)^2}{2\sigma^2}}R(a)da = \nabla_\mu F^\sigma(\mu)$, where $F^\sigma(\mu) = \int_a 1/(\sigma\sqrt{2\pi})e^{-\frac{(\mu-a)^2}{2\sigma^2}}R(a)da$ is a deterministic function given by passing the original reward landscape through a Gaussian filter. As shown in Figure 7, optimizing $R(a)$ with policy gradient will be equivalent to running gradient descent on $F^\sigma(\mu)$ with a sufficient number of samples. So if a non-convex reward $R(a)$ is still non-convex after it was smoothed by a Gaussian kernel, the random perturbation provided by the Gaussian noise will not help the policy to jump out of local optima.

We now consider the method starting with a very large standard deviation $\sigma$ and gradually reducing it to zero, as shown in Figure 7. We plot the $F^\sigma$ for different std $\sigma$. The leftmost shows $F^\sigma$ for large $\sigma$ and the rightmost is close to the original reward function. We use the red dotted line to denote the optimum $\mu^*(\sigma) = \arg\max_\mu F^\sigma(\mu)$ for each $\sigma$. We can see that for very large $\sigma$, $F^\sigma$ is convex, and we can assume the policy gradient method finds the optimal $\mu^*(\sigma)$. However, $\mu^*$ for very large $\sigma$ is not necessarily close to the global optima $\mu^*(0)$. In the beginning, the right mode is optimal as it has a high average reward (which can be visually measured by the area under the right mode). But when we reduce the $\sigma$ gradually, the global maximum may drastically change from the right mode to the left mode as shown in the red block (middle two) of Figure 7 because the left mode has a higher extreme value that exceeds the right. However, at the moment that the optimum changes to the left mode, $F^\sigma$ is already non-convex; the policy gradient method has little chance to discover the new global optima and will get stuck at the right local optima. This behavior will devastate all gradient-based algorithms that optimize a single-modal policy (REINFORCE, ADAM, SAC, and PPO).

To illustrate the insufficiency of the gaussian policy with a large standard deviation, we vary the initial standard deviation for REINFORCE and anneal to 0 in a sufficiently large amount of training steps. Notice that the global optima have a reward of 1.10.

| Initial std | 0.01 | 0.1 | 0.5 | 1 | 2 | 5 | 10 |
|---|---|---|---|---|---|---|---|
| expected reward | 0.50±0.00 | 0.50±0.00 | 0.50±0.00 | 0.50±0.00 | 0.49±0.00 | 0.47±0.00 | 0.42±0.00 |

Table 3: Final performance of reinforce with linearly decreasing standard deviation.

# I ABLATION STUDIES

In this section, we first study the importance of the trajectory encoder to understand what role it plays in helping maintain a multi-modality trajectory distribution. Then, we show how hyperparameter controlling the reward weight of entropy of action affects algorithms that have different policy paramterization.

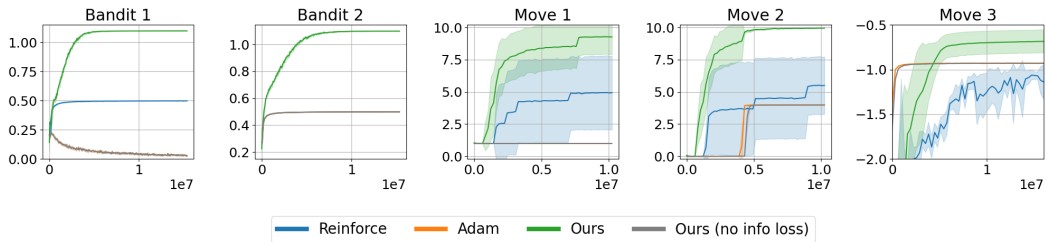

Figure 8: Ablating trajectory encoder

In Figure 8, we show the importance of the trajectory encoder, We can see that when we do not optimize the mutual information between $z$ and $\tau$ through $\log p_\phi(z|\tau)$ (Ours (no info loss)), our method degrades to almost identical behavior as Adam. We can see that the best-performing method is our method in its current form.

