# OpenReview forum: "Variational Reparametrized Policy Learning with Differentiable Physics"
_ICLR.cc/2023/Conference — Submitted to ICLR 2023_

### Official Review · Reviewer_5kNp · 2022-10-21

**Confidence:** 3
**Correctness:** 2
**Technical Novelty And Significance:** 2
**Empirical Novelty And Significance:** 2
**Recommendation:** 3

**Clarity, Quality, Novelty And Reproducibility:**

**Clarity/Novelty**: One of my main concerns is the submission's clarity. The proposed method is not properly motivated, the writing style and the paper's structure is poor and the paper's contributions are not delineated clearly enough from prior work. This makes it hard for the reader to grasp the central ideas and to judge the approaches validity. I encourage the authors to restructure their exposition and better guide the reader by better motivating the central theoretical insights.

**Quality**: As stated above, it is hard to judge the theoretical validity of the proposed approach due to the poor presentation. Furthermore, the paper focuses on the limited scope of differentiable environments and does not discuss nor empirically validate the applicability of VRP in the more general, non-differentiable RL setting. In addition, the empirical evaluation of VRP on differentiable environments is of low quality. I.p., the paper only contains results on a simple bandit problem and a grasping task and compares only against two very basic methods (REINFORCE, standard ADAM), completely ignoring any state of the art RL approaches (e.g., PPO or SAC). I encourage the authors to add more powerful baselines, extend their method to the non-differentiable setting and present quantitative evaluations on further problems.

**Reproducibility**: While the paper contains the hyperparameter settings used in the experiment, there is  no source code available (although the authors promise to provide it in the future). As the experimental settings are only vaguely described, it will be hard to reproduce the experiments from the paper alone.


**Strength And Weaknesses:**

Strengths: the basic idea of generatively modeling multimodal trajectory distributions is interesting.

Weaknesses:

- The structure and writing of the paper is poor, which makes it hard to follow.
- The paper only discusses the very limited setting of differentiable environments.
- The empirical evaluation is of low quality.

Please find detailed comments below.


**Summary Of The Paper:**

The paper studies reinforcement learning (RL) in high-dimensional continuous action spaces. It introduces Variational Reparametrized Policy (VRP), which formulates the policy as a generative model for optimal trajectories, optimized using a variational approach. The paper claims that this allows modeling complex and multi-model trajectory distributions. It evaluates VRP on two differentiable environments (a bandit and a grasping task) and compares against standard REINFORCE and Adam.


**Summary Of The Review:**

While generatively modeling multimodal trajectory distributions is an interesting idea, the submission lacks clarity and provides only very basic empirical evaluations with limited scope. I encourage the authors to improve the exposition of their work and extend the experimental evaluation. In its current form, I recommend to reject this submission.

---

> ### Author Response · Authors · 2022-11-19
> **Response to reviewer 5kNp**
>
> We sincerely thank you for your constructive feedback and valuable comments!
>
> > the paper focuses on the limited scope of differentiable environments and does not discuss nor empirically validate the applicability of VRP in the more general, non-differentiable RL setting.
>
> Our method can work when differentiable physics is not available.
>
> 1. The gradient of ELBO introduced in the method section can be easily estimated with a log-likelihood ratio estimator the same way as Reinforce does. This allows us to use our method in non-differentiable environments with no change to any other components.
> 2. The differentiable environment can be a learned neural network model. This is widely known and used in model-based RL.
>
> To illustrate it, we design a model-based algorithm in Appendix G and use the model learned during the exploration to solve multi-modal tasks with our method in non-differentiable environments.
>
> > The paper only compares only against two very basic methods (REINFORCE, standard ADAM), completely ignoring any state of the art RL approaches (e.g., PPO or SAC).
>
> Following your suggestion, we have added the comparison with PPO and SAC and added the results and detailed explanation in experiment section 4.1.2. Constrained by the single modality gaussian parameterization of the policy, these methods can only find the suboptimal solution, even when more than enough samples are given.
>
> > I encourage the authors to improve the exposition of their work and extend the experimental evaluation.
>
> We will make more thorough investigations and apply our methods on more challenging problems! We will also continue work on our writing to improve readability!

---

> > ### Comment · Reviewer_5kNp · 2022-12-02
> > **Thank you for your answer!**
> >
> > Thank you for your updates! I will take them into account in the discussion phase.

---

### Official Review · Reviewer_CRPH · 2022-10-24

**Confidence:** 4
**Correctness:** 2
**Technical Novelty And Significance:** 3
**Empirical Novelty And Significance:** 2
**Recommendation:** 3

**Clarity, Quality, Novelty And Reproducibility:**

**Quality**
While the theoretical derivations seemed fine, I was not convinced by the experimental work.
I think the work would have benefited from including some standard benchmarks, and showing results for a greater range of hyperparameters. Also ablation studies on the different components of the algorithm would have been useful.

Also, the literature discussion could have included works on variational skill discovery and option discovery.

**Clarity**
I think there was a lot of room for improving the writing.
The introduction could be more focused, and mention closely related works such as SeCTAR (Co-Reyes et al., ICML 2018).
It would also be good to clarify what exactly the contribution is.

I guess your method requires access to either a differentiable simulator or a model. These points could be made more explicit.

The discussion around discontinuities seems out of place. Also these issues were known much earlier than the cited works, e.g. see https://www.iro.umontreal.ca/~lecuyer/myftp/papers/intercha.pdf from 1993.

Levine's review was also not the first the mention control as inference, and it may have been good to cite some earlier works.

**Novelty**
There are several closely works, but I believe there are important differences to these related works.

**Reproducibility**
I believe sufficient details were given to reproduce the work.

**Strength And Weaknesses:**

**Strengths**
1. The method seemed like a technically sound way to learn a latent space for exploration.
2. The idea is widely applicable in reinforcement learning.

**Weaknesses**
1. The experimental comparisons all appeared new without any comparison on previous benchmarks or against previous implementations. Including at least one comparison with previous work might be good.
2. The theorems (primarily the gradient estimator and the discussion around discontinuities) directly follow from previous work, and I don't think there was anything to really derive. The proofs and regularity conditions appeared redundant to me as the continuity requirements are well-known.
3. I think the work lacked focus. For example, I think the main points of interest are whether the encoder is useful, and whether the ELBO objective is a useful theory. I didn't see any ablation study removing the encoder, or experiments where one samples a random $z$ giving this as an input to the policy and optimizing the rewards as usual. I would have liked to see stronger justifications regarding why the proposed method is useful in exactly the current embodiment compared to slight modifications of it.
4. The experiments appeared contrived to me. For example on the bandit 1 and 2 tasks, results for only one set of hyperparameters were shown. If the initial variance of the Gaussian distribution is set sufficiently high, I believe all of the methods should be able to solve these tasks. It would be good to test more hyperparameters and show the sensitivity of the methods.

**Summary Of The Paper:**

The work proposes a method for exploration in reinforcement learning.
The idea combines the concepts of variational inference and control as inference. In the method,
a variational autoencoder is used to learn a latent representation of the optimal trajectory distributions, where
the probability of optimality of a trajectory is proportional to $\exp(R(\tau))$ as in the typical control as inference
framework. The "decoder" in this framework includes the policy that takes the latent variable $z$ as an input and tries to maximize the rewards. The latent variable $z$ can either be sampled at the beginning of an episode and kept as a fixed input
to the policy, or it can also be periodically resampled using a state-dependent sampler.
It seems the main benefit is that $z$ leads to temporally correlated exploration.

One of the most closely related works is SeCTAR (Co-Reyes et al., ICML 2018) which uses a similar trajectory modeling framework, but
they use planning instead of policy search for action selection.
Other related works include skill or options learning frameworks that also use a latent variable input to the policy.
Another difference to prior works is that the current work uses differentiable simulation to aid in the optimization.

Experiments were performed in small bandit and other tasks, as well as in manipulation tasks with differentiable physics.

**Summary Of The Review:**

I recommend rejecting the paper.

While the idea was theoretically founded, I believe the writing lacked focus making the paper not that clear. Moreover, the experimental work did not appear thorough, and I believe it requires more investigation. Including comparisons with existing benchmarks or implementations would be good. Ablation studies on the different components of the algorithm would also aid in convincing me.

---

> ### Author Response · Authors · 2022-11-19
> **Response to reviewer CRPH (1/2)**
>
> We sincerely thank you for your timely review and thoughtful comments! We are glad that you found the idea interesting and theoretically sound.
>
> > Including at least one comparison with previous work might be good.
>
> We have added the comparison with PPO and SAC and added the results and detailed explanation in experiment section 4.1.2. Constrained by the single modality gaussian parameterization of the policy, these methods can only find suboptimal solutions, even when more than enough samples are given.
>
> > The theorems (primarily the gradient estimator and the discussion around discontinuities) directly follow from previous work, and I don't think there was anything to really derive. The proofs and regularity conditions appeared redundant to me as the continuity requirements are well-known.
>
> Thanks for pointing out the related literature [1], which is of great value! We have cited it in the revised manuscript. Following your suggestion, we moved the discussion on discontinuity to the appendix. We kept the proof in the appendix for completeness.
>
> We still want to clarify that our focus is to combine the likelihood ratio gradient estimate and the pathwise gradients in multi-modal policy learning instead of proving the theorem. As explained in the last paragraph of Sec 2.4, our form of theorem helps to solve the problem **even if the regularity condition does not hold in practice**. For example, for the bandit problem that involves discontinutities, our method can solve it while ADAM fails. Please see the animation ([website figure (1)](https://sites.google.com/view/iclr2023vrpl/home)) for more details.
>
> [1] L'Ecuyer, Pierre. "Note: On the interchange of derivative and expectation for likelihood ratio derivative estimators." Management Science 41.4 (1995): 738-747.
>
> > I didn't see any ablation study removing the encoder, or experiments where one samples a random z giving this as an input to the policy and optimizing the rewards as usual. I would have liked to see stronger justifications regarding why the proposed method is useful in exactly the current embodiment compared to slight modifications of it.
>
> We have updated the ablation study of removing the encoder to  Appendix I. We can see that when we do not optimize the mutual information between $z$ and $\tau$ through $\log p_\phi(z|\tau)$ (**Ours (no info loss)**) the method degrades to almost identical behavior as Adam.
>
> | Env      | Reinforce | Adam     | Ours     |  Ours (no info loss) |
> | -        | :-: | :-: | :-: | :-: |
> | Bandit 1 |0.50±0.00|0.03±0.01|**1.10**±0.00|0.03±0.01|
> | Bandit 2 |0.50±0.00|0.50±0.00|**1.10**±0.00|0.50±0.00|
> | Move 1   |4.93±2.85|1.00±0.00|**9.27**±1.36|1.00±0.00|
> | Move 2   |5.50±2.22|3.99±0.00|**9.95**±0.01|3.98±0.00|
> | Move 3   |-1.14±0.19|-0.93±0.00|**-0.68**±0.13|-0.93±0.00|
>
>
> > If the initial variance of the Gaussian distribution is set sufficiently high, I believe all of the methods should be able to solve these tasks.
>
> This is a very good question. **Even if the Gaussian distribution has a very high initial variance such that it can explore the whole space, it may still fail to solve these tasks**. The modality found by large initial standard deviation may not correspond to the global optima. Still, if we reduce the standard deviation, it will suffer from the issue of non-convexity.
>
> To illustrate the insufficiency of the gaussian policy with a large standard deviation, we vary the initial standard deviation for REINFORCE and anneal to 0 in a sufficiently large amount of training steps. Notice that the global optimum has a reward **$1.10$**.
>
> |Initial std| 0.01|0.1|0.5| 1|2|5|10|
> |:-: |:-:|:-:|:-:|:-:|:-:|:-:|:-:|
> |Gaussian |0.50±0.00|0.50±0.00|0.50±0.00|0.50±0.00|0.49±0.00|0.47±0.00|0.42±0.00|
>
> **We provide a detailed explanation for this phenomenon in Appendix H of our paper.**

---

> > ### Author Response · Authors · 2022-11-19
> > **Response to reviewer CRPH (2/2)**
> >
> > > Also, the literature discussion could have included works on variational skill discovery and option discovery; Levine's review was also not the first the mentioned control as inference, and it may have been good to cite some earlier works.
> >
> > We have included them in the revised manuscript. In particular, we added an additional section for variational skill discovery and extended discussion on hierarchical methods in section 3 (related work).
> >
> > We have also cited earlier work for RL as inference such as:
> > - Todorov, E. (2006). Linearly-solvable Markov decision problems
> > - Todorov, E. (2008). General duality between optimal control and estimation
> > - Toussant, M. (2009). Robot Trajectory Optimization Using Approximate Inference
> > - Ziebart, B. (2010). Modeling purposeful adaptive behavior with the principle of maximum causal entropy
> > - Kappen et. al. (2012). Optimal control as a graphical model inference problem
> >
> > Please let us know if we have missed any prior work on this topic.
> >
> > > I guess your method requires access to either a differentiable simulator or a model. These points could be made more explicit.
> >
> > Although our experiments focus on differentiable simulators,  it is easy to adapt our method to environments without them. We can either replace the pathwise gradient estimate with the standard RL methods or use a learned world model from the data. We design an illustrative algorithm to combine our approach with **learned models** in Appendix G. We also provide proof-of-concept experiments for non-differentiable environments to show its effectiveness.
> >
> > > While the idea was theoretically founded, I believe the writing lacked focus making the paper not that clear. Moreover, the experimental work did not appear thorough, and I believe it requires more investigation.
> >
> > Thanks for your appreciation! We will keep improving the paper presentation and investigate more thoroughly.

---

> > ### Comment · Reviewer_CRPH · 2022-11-21
> > **Thank you for the response**
> >
> > Thank you for taking my comments into consideration and updating the paper.
> >
> > I'm afraid that as Figure 4 is on page 10 (while the page limit is 9 pages), I will not be placing much weight on the update.
> >
> > I feel there is still too much discussion about the discontinuities. This does not seem to be the main focus of the method---the variational approach is not necessary to create methods that combine the reward-weighted and path-wise gradients. I would suggest to focus the discussion on explaining the advantage of using a variational approach.
> >
> > Regarding the experiments, it would help if a previously published result is replicated and compared to.
> >
> > The comparison with PPO and SAC on the toy tasks seemed unnecessary to me. But a comparison with PPO and SAC on more difficult robot control tasks would have helped.
> >
> > Another point that caught my attention is that Adam is an optimizer that can in principle be combined with the other approaches, e.g., REINFORCE can either use Adam or typical SGD. When comparing REINFORCE and pathwise gradients, the comparison should use the same optimizer.
> >
> > Also, it seems that the pathwise gradients will ignore most of the terms in $R_{elbo}$. This could be explained explicitly. Moreover, it may be good to place $R_{elbo}$ in an equation environment to make it more prominent. I think there are many similar modifications that could be made to improve the exposition.
> >
> > I encourage the authors to continue improving the experiments and the presentation of their work.

---

### Official Review · Reviewer_LHSH · 2022-10-25

**Confidence:** 3
**Clarity, Quality, Novelty And Reproducibility:** good
**Correctness:** 3
**Technical Novelty And Significance:** 3
**Empirical Novelty And Significance:** 3
**Recommendation:** 6

**Strength And Weaknesses:**

Strengths:
1) The proposed method is solid
2) The ability to leverage differentiable physics and mitigate local minima issue is desirable
3) Experiments shows the effectiveness of the algorithm

Weaknesses:
1) It’s not clear how well the method can work if a differentiable physics is not available.
2) It is mentioned that the method assumes lipschitz continuity in the problem while some of the presented tasks doesn’t seem to be so all the time (e.g. when the object make contact with the rope). How does the method handle this?
3) Furthermore, I'm wondering how well would the model be able to handle non-stable dynamics such as balancing a pole, or bipedal locomotion, where a small perturbation would lead to drastically different states later on.
4) The method seems to be also related to methods that uses mutual-information for encouraging diverse behaviors (e.g. Mutual Information State Intrinsic Control, Discovering Diverse Solutions in Deep Reinforcement Learning by Maximizing State-Action-Based Mutual Information, etc). Some discussions with these methods would be useful.


**Summary Of The Paper:**

The paper proposed a new RL algorithm for continuous control. A key idea is to represent the policy as a distribution of trajectories in the problem space to capture the multi-modality nature of the problem. A variational lower-bound is derived to create a practical optimization objective and with some assumptions on the smoothness of the problem a gradient calculation formulation is derived that can leverage differentiable physics simulation. The resulting method is applied to a few robotic control problems in simulation to show the effectiveness of the method.

**Summary Of The Review:**

The presented method is interesting and novel as far as I can tell.
As mentioned in the sections above, it'll be helpful if more discussions and potentially comparisons can be provided.

---

> ### Author Response · Authors · 2022-11-19
> **Response to reviewer LHSH (1/2)**
>
> We sincerely thank you for your appreciation! We are glad that you find our method solid and effective.
>
> > It’s not clear how well the method can work if a differentiable physics is not available.
>
> Our method can work when differentiable physics is not available. Besides optimizing the evidence lower bound directly with the zeroth-order gradient as in reinforcement learning, one can learn a differentiable world model [1, 2], which can also provide a good gradient estimate. To illustrate it, we design a model-based algorithm in **Appendix G** and use the model learned during the exploration to solve multi-modal tasks with our method.
>
>
> [1] Li, Yunzhu, et al. "Learning particle dynamics for manipulating rigid bodies, deformable objects, and fluids." arXiv preprint arXiv:1810.01566 (2018).
>
> [2] Hafner, Danijar, et al. "Dream to control: Learning behaviors by latent imagination." arXiv preprint arXiv:1912.01603 (2019).
>
>
> > It is mentioned that the method assumes Lipschitz continuity in the problem while some of the presented tasks don’t seem to be so all the time (e.g. when the object makes contact with the rope). How does the method handle this?
>
> This is a good observation. We want to clarify in advance that we only assume Lipschitz continuity when we need to use the pathwise gradient to optimize the policy. We do not need this assumption when we use RL or learned models as mentioned above.
>
> However, the reason why our method can handle tasks that involve discontinuous contacts is complex.
> - As mentioned in the last second paragraph of Sec 2.4, our method provides the potential to avoid discontinuities in the reward landscape by combining the sampling-based methods and the gradient-based optimization. Although the analytical gradient may not provide information to determine if the agent contacts with rope, the latent $z$ can still have the potential to model it by using certain, for example, $z_0$ to represent the modality with contacts and another $z_1$ to represent the modality without contacts. The reward-weighted gradient estimate selects the $z_0$ over $z_1$ based on its reward without being affected by the discontinuities. As an example, we visualize the optimization process ([website figure (1)](https://sites.google.com/view/iclr2023vrpl/home)) where you can see our method solves the task that involves discontinuities.
> - We also want to mention that many differentiable simulators apply different kinds of soft contacts [3,4,5] to facilitate optimization and avoid discontinuities.  For example, PlasticineLab applies a virtual force between two objects inversely proportional to their distance before they come into contact, making the environment much smoother. Besides, when we use the learned world model, the neural networks provide a continuous proxy of the original discontinuous environment. we illustrate it with the newly added model-based experiments in **Appendix G**.
>
> [3] Huang, Zhiao, et al. "Plasticinelab: A soft-body manipulation benchmark with differentiable physics." arXiv preprint arXiv:2104.03311 (2021).
>
> [4] Pang, Tao, et al. "Global Planning for Contact-Rich Manipulation via Local Smoothing of Quasi-dynamic Contact Models." arXiv preprint arXiv:2206.10787 (2022).
>
> [5] Werling, Keenon, et al. "Fast and feature-complete differentiable physics for articulated rigid bodies with contact." arXiv preprint arXiv:2103.16021 (2021).

---

> > ### Author Response · Authors · 2022-11-19
> > **Response to reviewer LHSH (2/2)**
> >
> > > I'm wondering how well would the model be able to handle non-stable dynamics such as balancing a pole, or bipedal locomotion, where a small perturbation would lead to drastically different states later on.
> >
> >
> > This is a profound question. To avoid a chaotic system, we include a finite horizon assumption. The Lipschitz continuity will then guarantee that the gradients and their expectations are bounded and exist. We can still use the pathwise gradient to estimate the true value gradient. The non-stable dynamics with large Lipscthiz only affect the estimate's variance (causing implicit bias in [6]) but will not hurt the optimality given large enough samples.
> > The visualization ([website figure(2)](https://sites.google.com/view/iclr2023vrpl/home)) on the website shows that our method based on gradients can easily solve the inverted double pendulum problem.
> >
> > For the infinite horizon case, pathwise gradients may not exist when the discount factor $\gamma$ is not able to mitigate exploding gradients. Determining the existence and the convergence of the value gradients becomes less non-trivial. In practice, there are several ways to handle long-horizon gradients. First, we can select a small enough $\gamma$ that is smaller than certain Lipscthiz factors of the dynamic system so that the gradients from the long-term future can be safely ignored. A second solution is to introduce a neural network to approximate the value function as in [7]. In **Appendix G**, we show that the gradient of a learned model, together with a Q network, can solve a bipedal locomotion task (Humanoid-v3), which is essentially non-differentiable.
> >
> > [6] Suh, Hyung Ju, et al. "Do differentiable simulators give better policy gradients?." International Conference on Machine Learning. PMLR, 2022.
> >
> > [7] Xu, Jie, et al. "Accelerated policy learning with parallel differentiable simulation." arXiv preprint arXiv:2204.07137 (2022).
> >
> >
> > >  Relations to methods that use mutual information for encouraging diverse behaviors
> >
> > Our method is closely related to mutual-information-based methods, which are used to discover diverse skills before. However, previous methods do not model the optimal trajectories and usually do not unify the trajectory optimization and the representation learning as ours.
> > Besides, our framework can support various forms of the latent variable $z$ and $p_\phi$ as a generalization of previous approaches. **We have included a new section in the related work to include your suggestion**.
> >
> > Welcome to add comments! Please do not hesitate to point out anything that we have missed!

---

### Decision · Program_Chairs · 2023-01-20

**Decision:**

Reject

**Justification For Why Not Higher Score:**

N/A

**Justification For Why Not Lower Score:**

N/A

**Metareview: Summary, Strengths And Weaknesses:**

This paper mainly considers the continuous space RL with known environment. The main contribution is to view the policy as a distribution of trajectories that allows the method to learn multi-modal policies.

According to the reviewers, there are mainly three main issues for this paper.  (1). The result is limited to problems with differentiable physics. Though the authors argue that one can learn the environment first and then apply the proposed method. However, learning the environment itself can be a another very hard problem, and it is not clear how accurate the learned model should be in order to make the proposed method work. (2). The theorems directly follow previous work, and the proofs and regularity conditions are well-known. This indicates that theoretically, the result lacks enough novelty. (3). The writing and presentation are poor, this makes it hard for the readers to follow the technical details of the paper.



**Summary Of Ac-Reviewer Meeting:**

N/A